# ON CAUSAL DISCOVERY IN THE PRESENCE OF DETERMINISTIC RELATIONS

## ABSTRACT

Many causal discovery methods typically rely on the assumption of independent noise, yet real-life situations often involve deterministic relationships. In these cases, observed variables are represented as deterministic functions of their parental variables, without noise. When determinism is present, constraint-based methods encounter challenges due to the violation of the faithfulness assumption. In this paper, we find, supported by both theoretical analysis and empirical evidence, that score-based methods with exact search can naturally address the issues of deterministic relations under rather mild assumptions. Nonetheless, exact score-based methods can be computationally expensive. To enhance the efficiency and scalability, we develop a novel framework for causal discovery that can detect and handle deterministic relations, called Determinism-aware Greedy Equivalent Search (DGES). DGES comprises three phases: (1) run Greedy Equivalent Search (GES) to obtain an initial graph, (2) identify deterministic clusters (i.e., variables with deterministic relationships), and (3) perform exact search exclusively on each deterministic cluster and its neighbors. The proposed DGES accommodates both linear and nonlinear causal relationships, as well as both continuous and discrete data types. Furthermore, we investigate the identifiability conditions of DGES. We conducted extensive experiments on both simulated and real-world datasets to show the efficacy of our proposed method.

## 1 INTRODUCTION

Causal discovery from observational data has attracted considerable attention in recent decades and has been widely applied in various fields such as machine learning (Nogueira et al., 2021), healthcare (Shen et al., 2020), manufacturing (Vuković & Thalmann, 2022) and neuroscience (Tu et al., 2019). Most causal discovery methods operate under the assumption of independent noises in the probabilistic system. However, real-world scenarios frequently encounter deterministic relationships. For example, the body mass index (BMI) is defined as the body weight divided by the square of the body height, composing a deterministic relation among weight, height, and BMI.

Constraint-based and score-based methods are two primary categories in causal discovery. Constraint-based methods, such as PC (Spirtes & Glymour, 1991) and FCI (Spirtes et al., 1995), leverage conditional independence tests (CIT) to estimate the graph skeleton and then determine the orientation. Under the Markov and faithfulness assumptions (Spirtes et al., 2000), these methods are guaranteed to asymptotically output the true Markov equivalence class (MEC). However, the faithfulness assumption is sensitive to many factors, such as the statistical errors with finite samples. Moreover, in the presence of deterministic relations, the faithfulness assumption is always violated. Take the chain structure $X \to Y \to Z$ for example where $Y = f(X)$. In this case, faithfulness is violated due to the conditional independence $Z \perp\!\!\!\perp Y|X$ or $Z \perp\!\!\!\perp f(X)|X$. Several variants of constraint-based methods (Ramsey et al., 2006; Spirtes & Zhang, 2014) have been proposed to allow certain types of unfaithfulness. However, they are generally not guaranteed to obtain the true MEC.

For score-based methods, the approach can vary based on the search strategy, which may involve greedy search, exact search, or continuous optimization. One typical score-based method with greedy search is Greedy Equivalent Search (GES) (Chickering, 2002), which searches in the space of MECs greedily by maximizing a well-defined score, such as Bayesian information criterion (BIC) score (Schwarz, 1978). Specifically, GES starts with an empty graph and consists of two phases.

In the forward phase, it incrementally adds one edge at a time if it yields the maximum score improvement, continuing until no further edge can be added to enhance the score. In the backward phase, it checks all edges to eliminate some if removal further improves the score. Similar to the aforementioned constraint-based methods, GES converges to the true MEC in the large sample limit.

Some exact score-based methods aim at weakening the faithfulness assumption required for asymptotic correctness of the search results, such as dynamic programming (DP) Koivisto & Sood (2004); Singh & Moore (2005), A* (Yuan et al., 2011; Yuan & Malone, 2013), and integer programming (Cussens, 2011; Bartlett & Cussens, 2017). The DAGs estimated by these methods can be converted to their MECs for causal interpretation (Spirtes & Zhang, 2018). Lu et al. (2021) demonstrated that these exact methods may produce correct results in cases where methods relying on faithfulness fail. Furthermore, Ng et al. (2021) proved that exact score-based search with BIC can asymptotically outputs the true MEC when the sparsest Markov representation (SMR) assumption (Raskutti & Uhler, 2018) is satisfied. Note that the SMR assumption is strictly weaker than the faithfulness assumption.

Deterministic relations have been considered in a few works of causal discovery. D-separation condition (Spirtes et al., 2000) is proposed for graphically determining conditional independence. Glymour (2007) proposed a heuristic procedure to learn the causal graph in a deterministic system, called DPC, where only a subset of variables will be conditioned in testing conditional independence. Daniusis et al. (2010) and Janzing et al. (2012) considered a deterministic system with only two variables, and presented the idea of independent changes to infer the causal direction. Luo (2006) and Lemeire et al. (2012) incorporated the classical PC algorithm and utilized additional independence tests to handle determinism. Mabrouk et al. (2014) combined a constraint-based approach with a greedy search that included specific rules to deterministic nodes and significantly reduce the incorrect learning. However, there is no identifiability guarantee in those related works Moreover, Zeng et al. (2021) assumes nonlinear additive noise model under high-dimensional deterministic data while Yang et al. (2022) assumes linear non-Gaussian model. Different from them, this paper aims to provide a principled framework to handle deterministic relations for arbitrary functional models. More related works are given in Appendix A2.

**Contributions.** Firstly, we find that exact score-based methods can naturally be used to address the issues of deterministic relations when mild assumptions are fulfilled. Secondly, due to the large search space of the possible DAGs, the exact score-based methods are feasible only for small graphs, therefore, we propose a novel framework called **D**eterminism-aware **G**reedy **E**quivalent **S**earch (DGES), aimed at enhancing the efficiency and scalability to handle deterministic relations. Importantly, DGES is a general three-phase method, with no restricted assumption on the underlying functional causal models, i.e., it can accommodate both linear and nonlinear relationships, Gaussian and non-Gaussian data distributions, as well as continuous and discrete data types. Thirdly, we provide the identifiability conditions of DGES under general functional models. Last but not least, we conducted extensive experiments on both simulated and real-world datasets to validate our theoretical findings and show the efficacy of our proposed method.

**Paper organization.** In Section 2, we review the common assumptions, provide a motivating example why PC fails in dealing with deterministic relations, then present our intuitive solution using exact score-based method. In Section 3, we present our proposed DGES with three phases in details. Furthermore, we provide the identifiability conditions for DGES presented in a general form in Section 4. The empirical studies in Section 5 validate our theoretical results and show the efficacy of our method. Finally, we conclude our work with further discussions in Section 6.

## 2 Causal Discovery with Deterministic Relations

In this section, we firstly review the preliminaries of causal discovery, especially with deterministic relations, then we provide some common assumptions that are related to our further analysis, as presented in section 2.1. Furthermore, we illustrate why using constraint-based methods such as PC algorithm can be problematic to address the deterministic issues, then we provide an intuitive solution to handle the issues by exact score-based methods, as shown in section 2.2.

### 2.1 Causal discovery and Common Assumptions

Let $\mathcal{G} = (\boldsymbol{V}, \boldsymbol{E})$ be a DAG with the vertex set $\boldsymbol{V}$ and edge set $\boldsymbol{E}$. Consider $d$ observable variables denoted by $\boldsymbol{V} = (V_1, V_2, ..., V_d)$, and denote $\mathbb{P}$ as its probability distribution. Given $n$ data samples,

the task of causal discovery aims at recovering the causal graph $\mathcal{G}$ from the data matrix $\boldsymbol{V} \in \mathbb{R}^{n \times d}$. Usually, each variable $V_i \in \boldsymbol{V}$ with random noises can be represented by the following structural causal model (SCM): $V_i = f_i(\text{PA}_i, \epsilon_i)$, where $\text{PA}_i$ is the set of all direct causes of $V_i$, and $\epsilon_i$ is the random noise with non-zero variance related to $V_i$, and we assume that $\epsilon_i$'s are mutually independent. For variables with deterministic relations, the SCM becomes: $V_i = f_i(\text{PA}_i)$, where there is no extra noise. Throughout this paper, we assume causal sufficiency, i.e., no latent confounder.

**Assumption 1 (Markov)** *Given a DAG $\mathcal{G}$ and the distribution $\mathbb{P}$ over the variable set $\boldsymbol{V}$, each variable is probabilistically independent of its non-descendants given its parents in $\mathcal{G}$.*

There are many DAGs which induce the same conditional independence relations with the distribution $\mathbb{P}$, and it is said to be Markov equivalent. The Markov equivalent class (MEC) contains all the DAGs which entail the same conditional independence relations as $\mathcal{G}$ does.

**Assumption 2 (Faithfulness (Spirtes et al., 2000))** *Given a DAG $\mathcal{G}$ and the distribution $\mathbb{P}$ over the variable set $\boldsymbol{V}$, $\mathbb{P}$ implies no extra CI relations which have not entailed by the Markov assumption.*

**Assumption 3 (Non-deterministic Faithfulness (Glymour, 2007))** *Given a DAG $\mathcal{G}$ and the distribution $\mathbb{P}$ over the variable set $\boldsymbol{V}$ which contains no deterministic variable, $\mathbb{P}$ implies no extra CI relations which have not entailed by the Markov assumption.*

Faithfulness is usually assumed for the whole set of variables, while non-deterministic faithfulness only focuses on the non-deterministic variables. When the Markov and faithfulness (Spirtes et al., 2000) assumptions hold true, constraint-based methods, such as PC, have been proved to asymptotically output the correct MEC. However, in the finite sample regime, the faithfulness assumption is sensitive to statistical testing errors when inferring the CI relations, and the violations might occur often. When there are deterministic relations, faithfulness also fails. Glymour (2007) proposes the non-deterministic faithfulness regarding only non-deterministic variables. Moreover, different relaxations of faithfulness have been proposed, such as adjacency-faithfulness (Ramsey et al., 2006) and triangle-faithfulness (Spirtes & Zhang, 2014). Another strictly weaker assumption is called Sparsest Markov Representation (SMR) (Raskutti & Uhler, 2018), which is also known as the unique-frugality assumption (Forster et al., 2018; Lam et al., 2022).

**Assumption 4 (Sparsest Markov Representation (SMR) (Raskutti & Uhler, 2018))** *Given a DAG $\mathcal{G}$ and the distribution $\mathbb{P}$ over the variable set $\boldsymbol{V}$, the MEC of $\mathcal{G}$ is the unique sparsest MEC which satisfies the Markov assumption.*

The idea behind the SMR is to find the most parsimonious or sparsest graphical representation that still captures the essential conditional independence relationships in the data. The term "sparsest" refers to the minimal number of edges in the graphical model. Under the SMR assumption, the exact score-based methods, such as the sparsest permutation (SP) and DP, have been shown to produce asymptotically correct results for learning the true MEC.

## 2.2 MOTIVATING EXAMPLE AND INTUITIVE SOLUTION

**Terminologies.** Consider the example with $d = 4$ variables, as shown in Figure 1, where $V_3$ has deterministic relation with $V_1$ and $V_2$ (i.e., $V_3 = V_1 + V_2$), and $V_4$ is a non-deterministic variable. Here we call the set of deterministic variables as a *deterministic cluster (DC)*, e.g., $\{V_1, V_2, V_3\}$ in Figure 1. Accordingly, all the non-deterministic variables make up a *non-deterministic cluster (NDC)*, e.g., $\{V_4\}$ in Figure 1. Meanwhile, the edges connecting between DC and NDC compose a *bridge set (BS)*, e.g., $\{V_2 \rightarrow V_4, V_3 \rightarrow V_4\}$ in Figure 1(a). Based on the above definitions and the motivating example in Figure 1, we provide the following lemma.

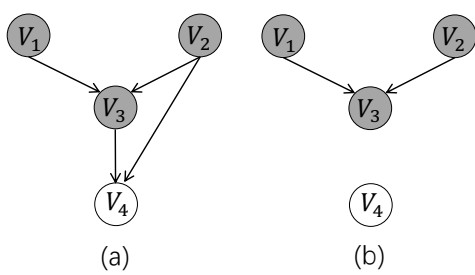

Figure 1: An example graph. (a) the true graph where $\{V_1, V_2, V_3\}$ is a deterministic cluster (DC). (b) the estimated graph by PC.

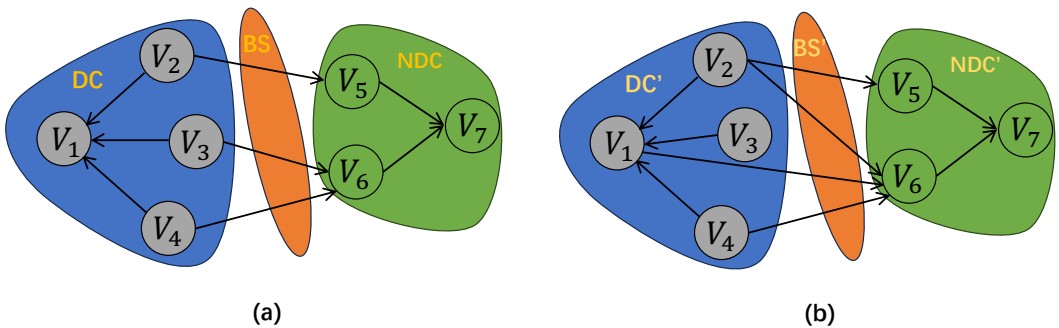

Figure 2: An example graph where $V_1 = f(V_2, V_3, V_4)$. (a) the true graph where DC = $\{V_1, V_2, V_3, V_4\}$, NDC = $\{V_5, V_6, V_7\}$, and BS = $\{V_2 \rightarrow V_5, V_3 \rightarrow V_6, V_4 \rightarrow V_6\}$. (b) the estimated graph by GES, where BS' = $\{V_2 \rightarrow V_5, V_1 \rightarrow V_6, V_2 \rightarrow V_6, V_4 \rightarrow V_6\}$

**Lemma 1 (Conditional Independence)** *Any deterministic variable, given the rest deterministic variables in DC, is always conditionally independent from any non-deterministic variable in NDC.*

*Basic idea of the proof.* Again take Figure 1 as the example. Given $\{V_1, V_2\}$, $V_3$ will always be conditionally independent from $V_4$, because $V_3$ can be perfectly determined by $\{V_1, V_2\}$ with no extra noise term, the estimated residue for regressing $V_3$ on $\{V_1, V_2\}$ will be close to 0. Therefore, $V_4 \perp\!\!\!\perp V_3 | \{V_1, V_2\}$ holds true, and similarly we have $V_4 \perp\!\!\!\perp V_2 | \{V_1, V_3\}$ and $V_4 \perp\!\!\!\perp V_1 | \{V_2, V_3\}$. That is why DNES is an empty set in Figure 1(b). The detailed proof is given in Appendix A3.

Under the umbrella of Lemma 1, the faithfulness assumption is violated as long as the deterministic relations are present. The key rule of constraint-based method (e.g., PC algorithm) is that: as long as we can find at least one conditional set or an empty set, so that two variables are independent or conditionally independent, then the edge between these two variables in the graph will be removed. Therefore, we can conclude that using constraint-based methods (e.g., PC algorithm) which rely on faithfulness to deal with the deterministic relations can be problematic.

However, benefiting from the recent theoretical progress on exact score-based methods, which do not explicitly rely on faithfulness assumption, it enables us to deal with deterministic relations from an intuitive view. Here, we are inspired by the lemma as follows.

**Lemma 2 (Linear Identifiability of Exact Search (Ng et al., 2021))** *Exact score-based search with BIC score asymptotically outputs a DAG that belongs to the MEC of the true DAG $\mathcal{G}$ if and only if the DAG $\mathcal{G}$ and distribution $\mathbb{P}$ satisfy the SMR assumption.*

According to Lemma 2, in the linear case, as long as the SMR assumption is satisfied, the exact score-based method with BIC score (Schwarz, 1978) can asymptotically obtain the true MEC. Then, we can extend the theoretical result from linear to nonlinear scenarios. The exact score-based method with generalized score (Huang et al., 2018) can also asymptotically output the true MEC.

**Theorem 3 (General Identifiability of Exact Search)** *Exact score-based search with generalized score asymptotically outputs a DAG that belongs to the MEC of the true DAG $\mathcal{G}$ if and only if the DAG $\mathcal{G}$ and distribution $\mathbb{P}$ satisfy the SMR assumption and some mild conditions are satisfied.*

**Remarks:** Based on the theoretical findings in Lemma 2 and Theorem 3, the exact score-based methods, which do not require faithfulness but SMR, pave a promising way to deal with the deterministic relations for causal discovery. However, one critical disadvantage of the exact methods is their low computational efficiency and poor scalability. To that end, we propose a novel framework, called DGES, which is demonstrated in section 3. The identifiability conditions for DGES are provided in section 4.

## 3 DETERMINISM-AWARE GREEDY EQUIVALENT SEARCH (DGES)

In this section, we will introduce our proposed DGES in detail. Throughout this paper, we consider the general case where some variables are deterministic while others are non-deterministic. There-

---

**Algorithm 1** DGES: **D**eterminism-aware **G**reedy **E**quivalent **S**earch

---

**Input:** data matrix $\mathcal{D} \in \mathbb{R}^{n \times d}$
**Output:** a causal graph $\mathcal{G}$
1: *(Phase 1: Run Greedy Search Globally)* Run greedy equivalent search on the whole set of variables to obtain an initial graph.
2: *(Phase 2: Detect Deterministic Clusters)* Detect the clusters of deterministic variables, by checking whether one variable can be perfectly represented by its neighbors.
3: *(Phase 3: Run Exact Search Partially)* Perform the exact search exclusively on the deterministic clusters and their neighboring variables.

---

fore, we can divide the whole causal graph into three parts: DCs, NDC, and BS. Regarding the BS part, in general, there are two scenarios. The cases, where NDC variables cause DC variables, can be trivially solved by GES, more details are given in Appendix A1. However, in the cases, where DC variables cause NDC variables, GES can be problematic. One example is given in Figure 2. Please note that although the graph in the example is directed, our method aims to identify the MEC, which may contain undirected edges as well.

Regarding NDC, the papers (Chickering, 2002; Huang et al., 2018) have shown the identifiability up to MEC of GES with BIC score or generalized score. As for DC, we cannot determine the edge directions without further assuming certain types of functional causal models, such as linear non-Gaussian model (Shimizu et al., 2006; Yang et al., 2022) and so on. Therefore, we focus on BS and aim to identify those sets of edges with mild assumptions, and we aim for a general framework where we do not assume any functional forms, as explained below.

## 3.1 GENERAL FRAMEWORK

In general, DGES contains three phases: Firstly, we run GES for the whole datasets, to get the initial causal graph. As mentioned above, GES has proved to be identifiable up to MEC with the non-deterministic relations, therefore, it is not necessary to run exact search for NDC part. Meanwhile, as demonstrated by Lu et al. (2021), GES may get sub-optimal results when the faithfulness assumption is violated, e.g., when there are deterministic relations. An example is given in Figure 2. Therefore, we need to partially conduct exact search based on the result of GES to accurately identify BS, which motivates the following phases. Secondly, we aim to detect the DCs and their neighbors. If one variable can be perfectly represented by its neighbors, we may conclude that it is a deterministic variable, and this variable together with its neighbors compose a DC. Thirdly, we perform the exact search exclusively on the DCs and their neighbors. The general framework is given in Algorithm 1.

## 3.2 SCORE FUNCTIONS

During the phase 1 with greedy search and phase 3 with exact search, a proper score function is inevitably needed. For any scoring criterion $\mathcal{S}(\mathcal{G}, \mathcal{D})$, we say that a score is *decomposable* if it can be written as a sum of local scores, where each local score is a function of only one variable and its parents. Following the property, the score of a DAG $\mathcal{G}$ can be represented as

$$\mathbb{S}(\mathcal{G}; \mathcal{D}) = \sum_{i=1}^{d} \mathcal{S}(V_i, \mathrm{PA}_i^{\mathcal{G}}). \tag{1}$$

Under the linear Gaussian model, the BIC score Schwarz (1978) is preferred, which is given as

$$\mathcal{S}_{BIC}(V_i, \mathrm{PA}_i^{\mathcal{G}}) = -\log L + \lambda' k \log n,$$
$$and \ \log L \propto -\frac{n}{2}(1 + \log |\Sigma|), \tag{2}$$

where $L$ is the maximized value of the likelihood function of the model based on the observed data $\mathcal{D}$ related to $V_i$ and $\mathrm{PA}_i$, $k$ denotes the number of edges between $V_i$ and $\mathrm{PA}_i$ in $\mathcal{G}$, $n$ is the number of data samples in $\mathcal{D}$, $\lambda'$ is the penalty parameter, $\Sigma$ is the variance of the noise term.

However, in the deterministic scenarios, the estimated noise variance $\hat{\Sigma}$ will asymptotically get closer to 0, which leads to numerical error because of the term $\log |\hat{\Sigma}|$. To deal with such an issue,

we provide the adjusted BIC score, formulated as

$$\mathcal{S}'_{BIC}(V_i, \mathrm{PA}_i^{\mathcal{G}}) = -\log L' + \lambda' k \log n,$$
$$and \ \log L' \propto -\frac{n}{2}(1 + \log|\Sigma + \xi|), \tag{3}$$

where $\xi$ is a small constant, and $\xi > 0$.

Under the general nonlinear model, the generalized score (GS) (Huang et al., 2018) which is in a non-parametric form is favored. There are two types of likelihoods as introduced in the paper, for computational efficiency, we choose the generalized score with cross-validated (CV) likelihood.

$$\mathcal{S}_{GS}(V_i, \mathrm{PA}_i^{\mathcal{G}}) = \frac{1}{Q}\sum_{q=1}^{Q}\ell(F_i^{(q)}|D_{0,i}^{(q)}), \quad and$$

$$\ell(\hat{\tilde{F}}_i^{(q)}|D_{0,i}^{(q)}) = -\frac{n_0^2}{2}\log(2\pi) - \frac{n_0}{2}\log\left|n_1\lambda^2\tilde{K}_{V_i}^{1(q)}(\tilde{K}_{PA_i^{\mathcal{G}}}^{1(q)} + n_1\lambda I)^{-2}\tilde{K}_{V_i}^{0(q)}\right|$$
$$-\frac{1}{2}\mathrm{trace}\left\{\frac{1}{\lambda}\tilde{K}_{V_i}^{0(q)}\tilde{K}_{V_i}^{0(q)} + \frac{1}{\lambda}\tilde{K}_{PA_i^{\mathcal{G}}}^{0,1(q)}A_i^{\mathrm{T}}A_i\tilde{K}_{PA_i^{\mathcal{G}}}^{1,0(q)} - n_1\tilde{K}_{PA_i^{\mathcal{G}}}^{0,1(q)}A_i^{\mathrm{T}}B_iA_i\tilde{K}_{PA_i^{\mathcal{G}}}^{1,0(q)}\right.$$
$$\left.+2n_1\tilde{K}_{V_i}^{0(q)}B_iA_i\tilde{K}_{PA_i^{\mathcal{G}}}^{1,0(q)} - \frac{2}{\lambda}\tilde{K}_{V_i}^{0(q)}A_i\tilde{K}_{PA_i^{\mathcal{G}}}^{1,0(q)} - n_1\tilde{K}_{V_i}^{0(q)}B_i\tilde{K}_{V_i}^{0(q)}\right\}, \tag{4}$$

where $A_i = \tilde{K}_{V_i}^{1(q)}(\tilde{K}_{PA_i^{\mathcal{G}}}^{1(q)} + n_1\lambda I)^{-1}$, $B_i = A_i(I + n_1\lambda A_i^{\mathrm{T}}A_i)^{-1}A_i^{\mathrm{T}}$, $\lambda$ is the regularization parameter, $n_1$ is the sample size of each training set, $n_0$ is the sample size of each test set, $n = n_1 + n_0$, $D_{1,i}^{(q)}$ and $D_{0,i}^{(q)}$ are the corresponding data of variable $V_i$ and its parents, $\tilde{K}_{V_i}^{1(q)}$ denotes the centralized kernel matrix of the $q$-th training set of $V_i$, $\tilde{K}_{V_i}^{0(q)}$ denotes that of the $q$-th test set of $V_i$, and similar notations are used for other kernel matrices.

### 3.3 DETECTING DETERMINISTIC CLUSTERS

**Terminologies.** In this section, we demonstrate how to detect the DCs. First of all, we would like to introduce some terminologies for analysis. A *root deterministic variable (RDV)* is the variable which can be perfectly represented by others in a DC. A *minimal deterministic cluster (MinDC)* refers to the variable set where there is only one RDV. A *maximal deterministic cluster (MaxDC)* refers to the union of all the DCs if there are multiple DCs in the graph. Note that in one causal graph, there are possibly multiple DCs while there should be just one NDC, we allow different DCs to share overlapping deterministic variables. Meanwhile, there could possily be multiple MinDCs but only one MaxDC. An *extended deterministic cluster (EDC)*, contains all the deterministic variables and their direct neighbors. For example in Figure 2(a), the RVD is $V_1$, both MinDC and MaxDC are $\{V_1, V_2, V_3, V_4\}$ since there is only one DC, and the EDC is $\{V_1, V_2, V_3, V_4, V_5, V_6\}$.

As shown in Figure 2(b), GES may output sub-optimal solutions where the estimated BS' contains more edges than the true one. Therefore, we need to rerun an exact search in order to recover the true BS under mild assumptions. Prior to running the exact search, it is essential to detect the true DCs and obtain their neighbors, i.e., the EDC. Based on the initial graph $\mathcal{G}_0$ given by GES in the first phase, we can infer that the deterministic variables in $\mathcal{G}_0$ will always present as a connected graph, though the edges may be different from the true ones. In other words, GES may output a sub-optimal solution $\mathcal{G}_0$, where the skeleton and directions related to the DC and BS can be different from these in the true graph $\mathcal{G}$, however, GES can always identify the correct set, which covers the true EDC. Therefore, it is reasonable and applicable to detect the true EDC by GES. Note that GES outputs a MEC which contains both directed and undirected edges. Furthermore, we provide the following theorem.

**Theorem 4 (Representation & Perfect Representation)** *Let $X$ be a random variable in the graph and $Z$ be the set of all direct causes and undirected neighbors of $X$, where $X$ and $Z$ are with domain $\mathcal{X}$ and $\mathcal{Z}$, respectively. Define a RKHS $\mathcal{H}_{\mathcal{X}}$ on $\mathcal{X}$ with continuous feature mapping $\phi_{\mathcal{X}} : \mathcal{X} \to \mathcal{H}_{\mathcal{X}}$.*

*Consider a regression framework in the RKHS: $\phi_\mathcal{X}(X) = F(Z) + u$, where $F : \mathcal{Z} \to \mathcal{H}_\mathcal{X}$ and $u$ represents the regression residue.*

*(i) $X$ can be represented by $Z$ if and only if*

$$\|\Sigma_u\|_{HS}^2 = 0, \tag{5}$$

*where $\Sigma_u$ is the variance matrix of the residue, $\Sigma_u = R_u^T R_u$, $R_u = \varepsilon(\boldsymbol{K}_Z + \varepsilon I)^{-1}\phi(X)$, $\varepsilon$ is a small positive regularization parameter for kernel ridge regression, and $\boldsymbol{K}_Z$ is the centralized kernel matrix of $Z$.*

*(ii) $X$ can be perfectly represented by $Z$, if and only if*

1) *$X$ can be represented by $Z$ and,*

2) *$X$ cannot be represented by any subset $Z_s$ of $Z$, where $0 < |Z_s| < |Z|$.*

**Remark.** If one variable $X$ can be perfectly represented by its direct causes and undirected neighbors $Z$, we can conclude that $X$ is a RDV, and $X$ and $Z$ forms a MinDC. Based on the above results, we can detect all the MinDCs and EDC, which is then fed into Phase 3 for exact search.

## 4 IDENTIFIABILITY CONDITIONS

In this section, we provide the identifiability conditions of DGES. The conditions are presented in a general form, applicable to both linear and nonlinear causal models. As mentioned above, in a general deterministic system, the whole causal graph mainly can be divided into three parts: DCs, NDC, and BS. In this paper, we focus on the identifiability for the BS and NDC parts.

**Theorem 5 (Partial Identifiability)** *Denote a causal graph $\mathcal{G}$ with deterministic relations. Let $V_i$ be any non-deterministic variable in $\mathcal{G}$, and $\mathrm{PA}_i$ be the set of direct causes or undirected neighbors of $V_i$ in one MinDC. Suppose the following conditions hold*

i. *Assumption 1, 3, and 4 hold,*

ii. *$|\mathrm{PA}_i| < |\mathrm{MinDC}| - 1$,*

*where $|\cdot|$ denotes the cardinality of a set. Then, when the sample size $n \to \infty$, the BS and NDC parts of the causal graph $\mathcal{G}$ is identifiable up to Markov equivalent class.*

## 5 EXPERIMENTS

To validate our theoretical findings and show the efficacy of our method, we conducted extensive experiments on simulated and real-world datasets. For simulated datasets, we evaluate linear and nonlinear causal models, as shown in Section 5.1. For real-world datasets, we evaluate a set of variables extracted from the pharmacokinetics dataset Grzegorzewski et al. (2021) in Section 5.2.

### 5.1 SIMULATED DATASETS

**Datasets and Implementations.** The true DAGs are simulated using the Erdös–Rényi model (Erdős et al., 1960) with the number of edges equal to the number of variables. The data is generated according to the functional causal model $V_i = \sum_{V_j \in \mathrm{PA}_i} b_{ij} f_i(V_j) + \epsilon_i$, where $V_j \in \mathrm{PA}_i$ is the $j$-th direct cause of $V_i$, $\epsilon_i$ is the random noise related to variable $V_i$, and $f_i$ is causal function. For deterministic variables, the noise term is removed, then the model becomes $V_i = \sum_{V_j \in \mathrm{PA}_i} b_{ij} f_i(V_j)$. For each setting, we randomly choose one RDV or two RDVs where each RDV has at least two parental nodes, in other words, there are one DC or two DCs in the generated graph. We evaluate both linear and nonlinear models. For linear Gaussian model, we let $f_i(V_j) = V_j$ and $\epsilon_i$ follows Gaussian distribution whose mean is zero and variance is uniformly sampled from $\mathcal{U}(1, 2)$. For nonlinear model, $f_i$ is randomly chosen from linear, square, sinc, and tanh functions, and $\epsilon_i$ is sampled from uniform distribution $\mathcal{U}(-0.5, 0.5)$ or Gaussian distribution $\mathcal{N}(0, 1)$. For the exact method in Phase 3, we choose A* (Yuan & Malone, 2013) without the heuristic tricks.

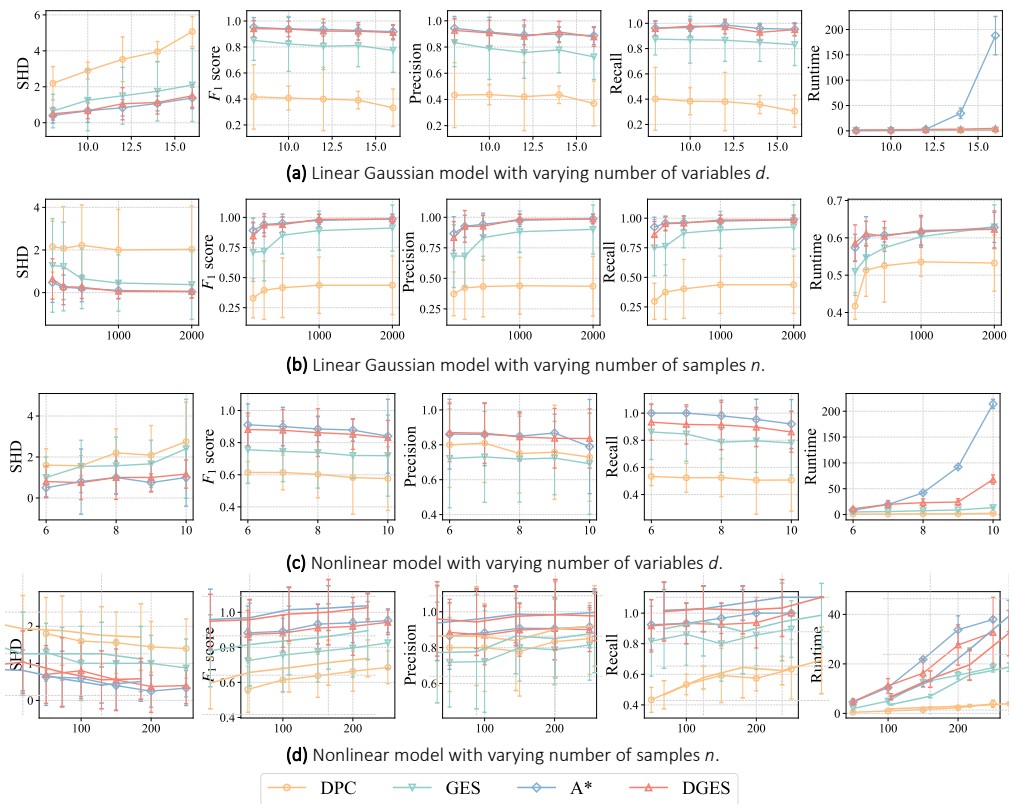

Figure 3: Simulated results on graph with one deterministic cluster. We evaluate different functional causal models on varying number of variables and samples, respectively. For each setting, we consider SHD ($\downarrow$), $F_1$ score ($\uparrow$), precision ($\uparrow$), recall ($\uparrow$) and runtime ($\downarrow$) as evaluation criteria.

**Baselines and Evaluations.** We compare our DGES with other baselines, including DPC Glymour (2007), GES Chickering (2002), and A* Yuan & Malone (2013). We compare the MEC of the output by all methods. For each method, we consider the structural Hamming distance (SHD), the $F_1$ score, the precision, the recall, and the computational time as evaluation criteria. Note that we only evaluate the BS part which we can identify in the graph under mild assumptions. We conduct the experiments on varying number of variables, varying number of samples, and some other hyperparameter studies. For linear model, we evaluate variable $d \in \{8, 10, 12, 14, 16\}$ while fixing sample size $n = 500$, and evaluate sample $n \in \{100, 250, 500, 1000, 2000\}$ while fixing variable $d = 8$. For nonlinear model, we evaluate variable $d \in \{6, 7, 8, 9, 10\}$ while fixing sample size $n = 100$, and evaluate sample $n \in \{50, 100, 150, 200, 250\}$ while fixing variable $d = 6$. We run 10 instances with different random seeds and report the means and standard deviations.

**Results and Analysis.** The simulated results about graphs with only one DC has been shown in Figure 4, and the results with two DCs (which may have overlapping variables) are given in Figure A3 of Appendix. Clearly, when there are more deterministic variables in the system, the runtime of our DGES will obviously increase. The reason is because there are more deterministic variables to be detected and fed into Phase 3 for exact search. According to the results, the general performance of DGES is competitive compared to other baselines. We observe that the exact method A* and our proposed DGES generally outperform the other baselines such as GES and PC across different criteria and settings. Meanwhile, score-based method GES presents better performance than constraint-based method DPC in a deterministic system. Accordingly, we can infer that score-based methods possess a holistic view for causal discovery, which may have less influence by the issues of deterministic relations, compared to constraint-based methods. As the number of variable increasing, the runtime of A* will increase rapidly. Compared to A*, the increasing of runtime for DGES is much more steady, both in linear and nonlinear models. Due to the space limit, the implementation details and the experimental results with two DCs are provided in Appendix A4.

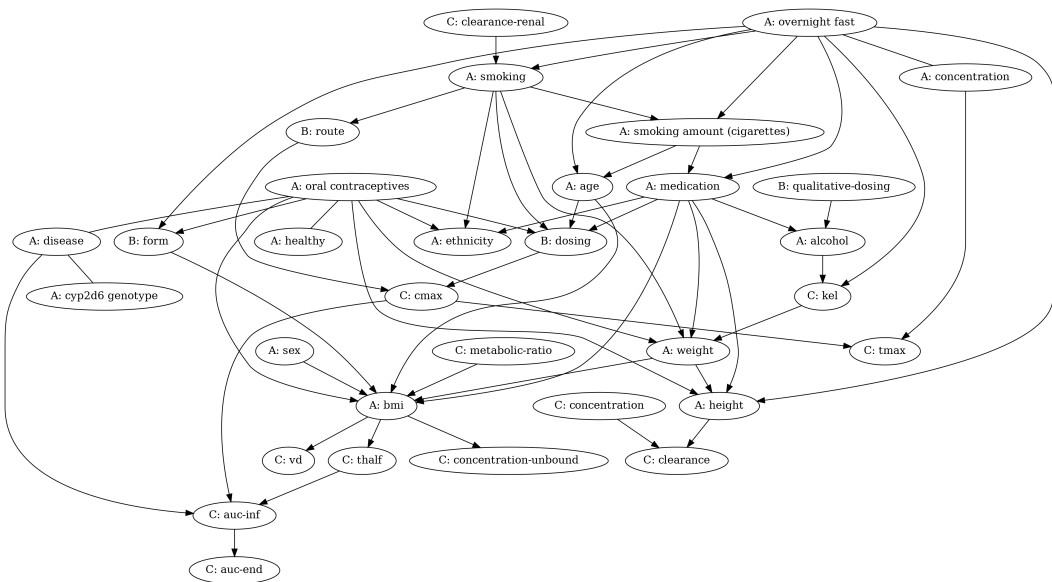

Figure 4: Results on real-world dataset with deterministic relations by our proposed method.

## 5.2 REAL-WORLD DATASET

We also evaluate our method and the baselines on a real-world dataset, namely the pharmacokinetics dataset Grzegorzewski et al. (2021), which is an open database for pharmacokinetics information from clinical trials. It provides curated information mainly in three categories: the characteristics of the studied patients (e.g., age, height), applied records of the studied drugs (e.g., the dosing of drug, the route of application), and the measurement records (e.g., the clearance, $T_{max}$, $C_{max}$ when one certain person takes one certain drug), and we name the three categories of variables as class "A", "B", and "C", respectively. Out of more than 200 variables and more than 200000 data samples containing missing values, we cleaned the data and finally obtained 32 important variables with 4194 data samples which may contain deterministic relations. The 32 variables contains 16, 4, and 12 variables from the class "A", "B" and "C", respectively. Considering the rather large number of graph and large sample size, we use linear BIC score instead of nonlinear generalized score to conduct the search. As shown in Figure 4, our method can successfully detect at least three deterministic clusters: {height, weight, BMI}, {$T_{max}$, $C_{max}$}, {$AUC_{inf}$, $AUC_{end}$}.

## 6 DISCUSSION AND CONCLUSION

**Discussions.** While presenting a versatile framework, our paper does have certain limitations. First of all, our work provides a general framework under mild assumptions, which cannot identify the skeleton and directions in the DC part so far. Without further assumptions such as the linear non-Gaussian causal model, current information is not enough to identify the DC part. More discussion is in Appendix A1. Furthermore, inherited from the disadvantages of exact methods, our method also can be computationally slow in Phase 3 when there are a large number of deterministic variables.

**Conclusion.** This paper dives into the challenges of causal discovery in the presence of deterministic relations. Notably, we make a compelling discovery that exact score-based methods can elegantly address the deterministic issues, provided the SMR assumption is met. In an effort to bolster efficiency and scalability in a deterministic system, we propose the novel and versatile framework called DGES, encompassing both linear and nonlinear models, as well as both continuous and discrete data types. Furthermore, we establish the identifiability conditions for DGES, which is easy to be satisfied. Hopefully, our method can help to construct a holistic view to see the deterministic relations. The extensive experiments, conducted on both synthetic and real-world datasets, validate our theoretical findings and the efficacy of our method. In the future, we will enhance our method to achieve the fully identifiability among the deterministic variables, by incorporating more assumptions.

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

*Appendix for*

## "On Causal Discovery in the Presence of Deterministic Relations"

Appendix organization:

## A1  MORE DISCUSSIONS

**Q1: Why current method cannot identify the skeleton and directions in the DC part?**

**A1:** To achieve that goal, we usually need strong assumptions on the underlying functional causal model, i.e., Yang et al. (2022) assumed linear Non-Gaussian model. However, those assumptions are not in alignment with our goal of a general method, i.e., with no restricted assumption on the underlying functional causal model. That is why currently our method cannot identify the skeleton and directions in the DC part. However, fortunately, we can exactly find out which set of variables are in the DCs using some DC detection strategies, as shown in Section 3.3.

**Q2: Why the cases, where NDC variables cause DC variables, can be trivially solved by GES?**

**A2:** Let's consider this example. Three variables compose a DC ($V_1$, $V_2$, $V_3$, and $V_1 = V_2 + V_3$), and denote another variable from NDC as $V_4$. In this case, there must not be an edge from $V_4$ to $V_1$, because $V_4$ will be in DC rather than NDC, if so. Then, if $V_4$ causes $V_2$, there will be definitely an edge between them by GES, because $V_4$ is clearly dependent on $V_2$ and theoretically GES can capture this dependence based on the local consistency.

**Q3: Why GES can be problematic in the cases where DC variables cause NDC variables?**

**A3:** Take Figure 2 as an example, where the true edges related to $V_6$ include $V_3 \rightarrow V_6$ and $V_4 \rightarrow V_6$. However, during the forward phase of GES, it is very likely that the edge $V_1 \rightarrow V_6$ can be added in the beginning. Then the edges $V_2 \rightarrow V_6$ and $V_4 \rightarrow V_6$ are added subsequently. During the backward phase of GES, the edge $V_1 \rightarrow V_6$ will not be deleted, because $V_1$ also contains information from $V_3$, in other words, $V_6$ is represented by $V_1, V_2$ and $V_4$ by GES, which contains more edges than the ground-true. Therefore, in this case GES can be problematic, and we need exact search under the SMR assumption for post-processing, in order to correctly identify the BS part.

**Q4: What the characterization of Markov equivalence class is in the context of having deterministic relations in the whole graph?**

**A4:** Regarding only the variables involved in BS and NDC (that is how Theorem 5 claimed), the characterization of Markov equivalence class (MEC) is still the same in the context of having deterministic relations or not having such relations.

However, if we consider the whole graph, i.e., all of the variables in DCs are also involved, the characterization of the Markov equivalence class should be different. As shown in the example of Lemma 1 and Figure 1, there will be "spurious CI statements" caused by the deterministic relations, therefore, we need to remove those "spurious CI statements" for the new characterization of MEC.

**Q5: Why use kernel regression in Theorem 4?**

**A5:** The reason why we use kernel regression is that: we are considering the general functional causal form, particularly, this theorem can be used for both linear and nonlinear functional relationships, Gaussian and non-Gaussian data distributions, which is in alignment with the general goal of our proposed method.

For more details, inspired by (Zhang et al., 2012), the functions $\phi_{\mathcal{X}}$ and $F(\cdot)$ that we use are all in the infinite Hilbert spaces, and we evaluate the representation with the Hilbert-Schmidt norm of the variance operator $\Sigma_u$ in infinite dimension. In this case, we can exhibit a general functional causal form.

## A2 RELATED WORKS

In this part, we will introduce more related works in causal discovery (Spirtes & Zhang, 2016). As we mentioned in the main paper, constraint-based and score-based methods are two primary categories in causal discovery. Constraint-based methods utilize the conditional independence test (CIT) to learn a skeleton of the directed acyclic graph (DAG), and then orient the edges upon the skeleton. Such methods contain Peter-Clark (PC) algorithm (Spirtes & Zhang, 2016) and Fast Causal Inference (FCI) algorithm (Spirtes, 2001). Some typical CIT methods include kernel-based independent conditional test (Zhang et al., 2012) and approximate kernel-based conditional independent test (Strobl et al., 2019).

Score-based methods normally use a score function and rely on a particular search strategy to look for the intended graph. The search strategy usually involve greedy search, exact search, or continuous optimization. The first continuous-optimization based method is NOTEARS (Zheng et al., 2018), which casts the Bayesian network structure learning task into a continuous constrained optimization problem with the least squares objective, using an algebraic characterization of directed acyclic graph (DAG). Subsequent work GOLEM (Ng et al., 2020) adopts a continuous unconstrained optimization formulation with a likelihood-based objective. NOTEARS is designed under the assumption of the linear relations between variables, therefore, another subsequent works have extended NOTEARS to handle nonlinear cases via deep neural networks, such as DAG-GNN (Yu et al., 2019) and DAG-NoCurl (Yu et al., 2021). Moreover, ENCO (Lippe et al., 2022) presents an efficient DAG discovery method for directed acyclic causal graphs utilizing both observational and interventional data. AVCI (Lorch et al., 2022) infers causal structure by performing amortized variational inference over an arbitrary data-generating distribution. These methods might suffer from various optimization issues, including convergence (Wei et al., 2020), sensitivity to data standardization (Reisach et al., 2021), and nonconvexity (Ng et al., 2023). Since they are only guaranteed to find a local optimum, therefore the quality of the solution can not be guaranteed, even in the asymptotic cases.

Besides the constrain-based and score-based methods, another major category of causal discovery methods is function causal model based methods. Those methods rely on the causal asymmetry property, such as the linear non-Gaussian model (LiNGAM) (Shimizu et al., 2006), the additive noise model (Hoyer et al., 2008), and the post-nonlinear causal model (Zhang & Hyvarinen, 2012). Apart from those methods, there are also some hybrid methods, such as neural conditional dependence (NCD) method, which reframes the GES algorithm to be more flexible than the standard score-based version and readily lends itself to the nonparametric setting with a general measure of conditional dependence.

## A3 PROOFS

In this section, we provide the proofs of theorems and lemmas in the main paper, including Lemma 1, Theorem 3, Theorem 4, and Theorem 5.

### A3.1 PROOF OF LEMMA 1

**Proof:** Assume a deterministic cluster $S = \{X\} \cup Y$, where $X$ is any one variable in the DC, and $Y$ is the set of the other deterministic variables in S.

By the definition of determinisic relation, we can have

$$f(X, Z) = 0, \tag{6}$$

In other words, we may also obtain $X = g(Z)$ or $Z = h(X)$ without a random noise term.

When we are doing conditional independent test to evaluate the null hypothesis $Y \perp\!\!\!\perp X|Z$, where $Y$ is any one non-deterministic variable in the system, then a standard procedure is to conduct regression in RKHS (Zhang et al., 2012; Huang et al., 2018) in the following form

$$\begin{aligned} \phi_X &= F_1(Z) + u_1, \\ \phi_Y &= F_2(Z) + u_2, \end{aligned} \tag{7}$$

where $\phi$, $F_1$ and $F_2$ are the nonlinear feature mapping in the RHKS.

Then the null hypothesis $Y \perp\!\!\!\perp X|Z$ holds true, if and only if the $\|\Sigma_u\|_{HS}^2 = 0$ where $\Sigma_u$ is the variance matrix of two residues $u_1$ and $u_2$.

Given that $X$ and $Z$ are deterministically related, and also $X = g(Z)$. Therefore, the residue term after the kernel ridge regression will always be $\mathbf{0}$ (Perfectly representation! Note that $X$ corresponds to $\phi_X$ and $g(Z)$ corresponds to $F_1(Z)$ in the Eq. 7).

In other words, $\|\Sigma_u\|_{HS}^2 = 0$ will always holds true. Then, the null hypothesis will also hold true: $Y \perp\!\!\!\perp X|Z$.

To summarize, we can conclude: $Y \perp\!\!\!\perp X|Z$ will always hold true, if $X$ and $Z$ deterministically related. To extend this result from one variable $Y$ to arbitrary non-deterministic variable, and extend the result from one variable $X$ to arbitrary deterministic variable. We can conclude that: Any deterministic variable, given the rest deterministic variables in DC, is always conditionally independent from any non-deterministic variable in NDC.

Proof ends.

### A3.2 PROOF OF THEOREM 3

**Proof:** As suggested by the generalized score (Huang et al., 2018), with proper score functions and seach procedures, asymptotically the resulting Markov equivalence class has the same independence constraints as the data generative distribution.

(i) First of all, we would like to discuss the local consistency of generalized score.

For the regression problem, one can define the effective dimension of the kernel space and the complexity of the regression function according to Caponnetto & De Vito (2007). Then under mild conditions, the CV-likelihood score is locally consistent.

**Lemma 6** *Suppose that the sample size of each test set $n_0$ satisfies*

$$n_0 \to \infty, \frac{n_0}{n} \to 0 \text{ as } n \to \infty,$$

*and suppose that the regularization parameter $\lambda$ satisfies*

$$\lambda = O(n^{-\frac{b}{bc+1}}),$$

*where $n$ is the total sample size, $b$ is a parameter of the effective dimension of the kernel space with $b > 1$, and $c$ indicates the complexity of the regression function with $1 < c \le 2$.*

**Lemma 7** *Assume that all conditions given in Lemma 6 hold. With the CV likelihood under the regression framework in RKHS as a score function and with the GES search procedure, it guarantees to find the Markov equivalence class which is consistent to the data generative distribution asymptotically.*

Lemma 7 ensures that, with proper score functions and seach procedures, asymptotically the resulting Markov equivalence class has the same independence constraints as the data generative distribution. For the complete proofs, please refer to the Appendix A5 of paper (Huang et al., 2018).

(ii) Then, We will provide the proof by contra-positive in both directions based on the consistency of the generalized score as shown above.

1) "If" direction:

Suppose that exact score-based search asymptotically outputs a DAG $\mathcal{H}$ (having the highest generalized score) that does not belong to the MEC of the true DAG $\mathcal{G}$. Since the generalized score is known to be consistent, $(\mathcal{H}, \mathbb{P})$ must satisfy the Markov assumption, because otherwise its generalized score is lower than that of the true DAG $\mathcal{G}$ and exact search would not have output $\mathcal{H}$. By assumption, the generalized score of $\mathcal{H}$ is higher than that of $\mathcal{G}$, which, by the consistency of generalized, implies that $|\mathcal{H}| \le |\mathcal{G}|$, and therefore, $(\mathcal{G}, \mathbb{P})$ does not satisfy the SMR assumption.

2) "Only if" direction:
Suppose that $(\mathcal{G}, \mathbb{P})$ does not satisfy the SMR assumption. Then there exists a DAG $\mathcal{H}$ not in the MEC of $\mathcal{G}$ such that $|\mathcal{H}| \le |\mathcal{G}|$, and $(\mathcal{H}, \mathbb{P})$ satisfies the Markov assumption. Without loss of generality, we choose $\mathcal{H}$ with the least number of edges. We first consider the case in which $|\mathcal{H}| < |\mathcal{G}|$. Since both $\mathcal{H}$ and $\mathcal{G}$ satisfy the Markov assumption, by the consistency of generalized, the generalized score of $\mathcal{H}$ is higher than that of $\mathcal{G}$, which implies that exact score-based search will not output any DAG from the MEC of $\mathcal{G}$. For the case with $|\mathcal{H}| = |\mathcal{G}|$, since they are both Markov with distribution $\mathbb{P}$, they have the same generalized score. Therefore, exact search will output a DAG that belongs to the MEC of either $\mathcal{H}$ or $\mathcal{G}$, and is not guaranteed to output a DAG from the MEC of the true DAG $\mathcal{G}$.

Proof ends.

### A3.3    PROOF OF THEOREM 4

**Proof:** We will divide the whole proofs into two parts. For the first part, we aim to prove the "representation" theorem, and for the second part, we aim to further prove the "perfect representation" theorem.

(i) Representation:

Assume there is a MEC $\mathcal{M}$, which contains both directed edges and undirected edges. Let $X$ be a random variable in $\mathcal{M}$ and $Z$ be the set of all non-descendant neighbors including direct causes and undirected neighbors of $X$. Suppose the random variables $X$ and $Z$ are over measureable spaces $\mathcal{X}$ and $\mathcal{Z}$, respectively.

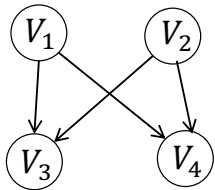

Figure A1: An example graph with deterministic relations: $V_3 = f(V_1, V_2), V_4 = g(V_1, V_2)$.

Without assuming a particular functional causal form, we usually exploit a regression framework in the RKHS, to encode general dependence relations between two random variables.

Define a RKHS $\mathcal{H}_\mathcal{X}$ on $\mathcal{X}$ with continuous feature mapping $\phi_\mathcal{X} : \mathcal{X} \to \mathcal{H}_\mathcal{X}$. Here, we consider

$$\phi_\mathcal{X}(X) = F(Z) + u, \tag{8}$$

where $F : \mathcal{Z} \to \mathcal{H}_\mathcal{X}$ and $u$ represents the regression residue or noise. When applying the kernel ridge regression, we can obtain the estimated residue

$$\hat{u} = \varepsilon(\boldsymbol{K}_Z + \varepsilon I)^{-1}\phi(X), \tag{9}$$

where $\varepsilon$ is a small positive regularization parameter for kernel ridge regression, and $\boldsymbol{K}_Z$ is the centralized kernel matrix of $Z$. To evaluate whether such a residue exists, one may consider Hilbert-Schmidt norm of the variance matrix

$$\Sigma_{\hat{u}} = \hat{u}^T \hat{u} = 0, \tag{10}$$

If the above equation holds true, then we may conclude that there is no noise term in the relationship between $X$ and $Z$, in other words, $X$ can be represented by $Z$ (without extra noise term).

Vice versa.

(ii) Perfect representation:

Intuitively speaking, the perfect representation can be motivated by the example, as shown in Figure A1, where $V_3 = f(V_1, V_2)$, $V_4 = g(V_1, V_2)$.

When we are representing the variable $V_3$, there should be multiple ways

$$\begin{aligned}
V_3 &= f(V_1, V_2) \\
&= h(V_1, V_2) + g(V_1, V_2) \\
&= h(V_1, V_2) + V_4.
\end{aligned} \tag{11}$$

Accordingly, we can say that $V_3$ can be represented by $\{V_1, V_2\}$ or $\{V_1, V_2, V_4\}$. Clearly, the minimum deterministic cluster should be $\{V_1, V_2, V_3\}$ and $\{V_1, V_2, V_4\}$. In other words, $\{V_1, V_2, V_4\}$ can be redundant in the relationships to represent $V_3$. Therefore, we can conclude that: $V_3$ can be represented by $\{V_1, V_2, V_4\}$, but perfectly represented by $\{V_1, V_2\}$.

Here, if we cannot find a smaller subset of current set for representation, then we can say that this current set is a perfect representation of that variable. Mathematically, we have:

$X$ can be perfectly represented by $Z$, if and only if

   1) $X$ can be represented by $Z$ and,
   2) $X$ cannot be represented by any subset $Z_s$ of $Z$, where $0 < |Z_s| < |Z|$.

Vice versa. With the concept of the perfect representation, we can easily detect all the MinDC in our graph.

Proof ends.

## A3.4    PROOF OF THEOREM 5

**Proof:**

First of all, we will explain why we need the listed three assumptions. Then, we will explain why we need to have constraint on $|\,\mathrm{PA}_i\,| < |\,\mathrm{MinDC}\,| - 1$.

(i) As mentioned in our main paper, there are three phases of our proposed DGES. During the first phase, we need to run GES. To ensure the accuracy of output (particularly on the NDC part), we

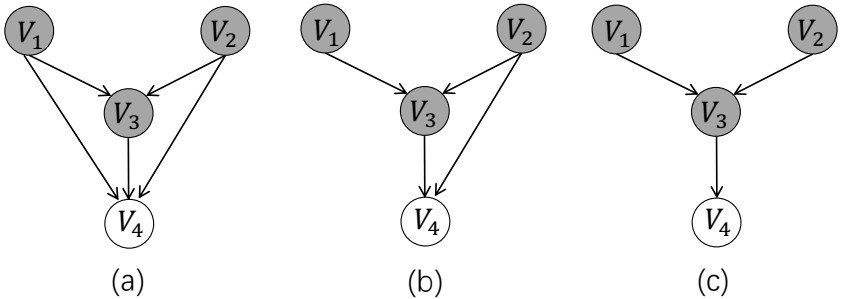

Figure A2: An example graph with determinisic relation where $V_3 = f(V_1, V_2)$. (a) A non-deterministic variable $V_4$ connects to $\{V_1, V_2, V_3\}$. (b) A non-deterministic variable $V_4$ connects to $\{V_2, V_3\}$. (c) A non-deterministic variable $V_4$ connects to $\{V_3\}$. Here among the three graphs, only the graph (c) can be partially identified.

need the assumptions of Markov and non-deterministic faithfulness (See Assumption 1 and 3). Then in the third phase, we need to perform the exact search exclusively on the EDC, where the Sparsest Markov Representation (SMR) assumption will be needed.

(ii) As for why we need to condition on $|\operatorname{PA}_i| < |\operatorname{MinDC}| - 1$, we can start with explaining why $|\operatorname{PA}_i| = |\operatorname{MinDC}|$ and $|\operatorname{PA}_i| = |\operatorname{MinDC}| - 1$ will fail the provided identifiability.

Let's take an example with four variables, where three of them are deterministically related, as shown in Figure A2. Here among the three graphs, only the graph (c) can be partially identified, and the graph (a) and (b) cannot achieve partial identifiability.

We further assume a linear functional causal model, then we can formulate the deterministic relationship as

$$aV_1 + bV_2 + cV_3 = 0, \tag{12}$$

where $a, b, c$ are any linear coefficients. Based on the above formulation, the causal equation of variable $V_4$ in Figure A2(a) can be represented as

$$
\begin{aligned}
V_4 &= dV_1 + eV_2 + fV_3 + \epsilon \\
&= dV_1 + eV_2 + f\frac{1}{c}(aV_1 + bV_2) + \epsilon \\
&= dV_1 + e\frac{1}{b}(aV_1 + cV_3) + fV_3 + \epsilon \\
&= d\frac{1}{a}(bV_2 + cV_3) + eV_2 + fV_3 + \epsilon,
\end{aligned}
\tag{13}
$$

where $\epsilon$ is the random noise injected into $V_4$. Clearly, the above four equations are all valid, in other words, $V_4$ can be possibly represented by different sets of variables, meaning that this case is not guaranteed to be identified.

Regarding the variable $V_4$ in Figure A2(b), the causal equation can be represented as

$$
\begin{aligned}
V_4 &= eV_2 + fV_3 + \epsilon \\
&= eV_2 + f\frac{1}{c}(aV_1 + bV_2) + \epsilon \\
&= e\frac{1}{b}(aV_1 + cV_3) + fV_3 + \epsilon.
\end{aligned}
\tag{14}
$$

Again, the above three equations are all valid, in other words, $V_4$ can be possibly represented by different sets of variables, meaning that this case is also not guaranteed to be identified.

However, in Figure A2(c), things are different. The causal equation of variable $V_4$ can be represented as

$$
\begin{aligned}
V_4 &= fV_3 + \epsilon \\
&= f\frac{1}{c}(aV_1 + bV_2) + \epsilon.
\end{aligned} \tag{15}
$$

When the SMR assumption is satisfied, we can identify the only one case, which is $V_3 \to V_4$.

Now, we extend the three-variable case to the general linear case where there is a MinDC with the cardinality $|\operatorname{MinDC}|$. And we can easily conclude the true conditions to be: $|\operatorname{PA}_i| < |\operatorname{MinDC}| - 1$.

Furthermore, we extend the linear to nonlinear case, where we can also conclude that the listed conditions ensures the partial identifiability.

Proof ends.

## A4   MORE DETAILS ABOUT THE SIMULATED EXPERIMENTS

### A4.1   IMPLEMENTATION DETAILS

We provide the implementation details of our method and other baseline methods for synthetic datasets.

- DPC (Glymour, 2007): The method is an extension for traditional PC algorithm (Spirtes et al., 2000), the key idea is that: every time when we do the conditional independence test, we aim to remove the potential deterministic variables from the conditioning set so that the faithfulness will not be violated. Here we follow the paper, and use the covariance to measure the closeness of two variables. If the covariance between two variables are greater than 0.9, we then remove the variable from the conditioning set in conditional independence test. Meanwhile, for linear Gaussian model, we choose FisherZ test, while for nonlinear model we choose kernel-based test (Zhang et al., 2012), and the significance level is set to $\alpha = 0.05$ by default. We implement this method based on the Causal-learn package `https://github.com/py-why/causal-learn` (Zheng et al., 2023).

- GES (Chickering, 2002): This method is a classical score-based method with greedy search. Our implementation is based on the code from `https://github.com/juangamella/ges`. For linear Gaussian model, we use BIC score. And for general nonlinear model, we use generalized score with cross-validation likelihood (Huang et al., 2018). The penalty parameter for controlling the sparsity is set to 1.

- A* (Yuan & Malone, 2013): A* is one of the classical exact score-based methods. Actually, there are some heuristic algorithms proposed to accelerate the search procedure. Considering in our scenarios, we do not utilize any heuristic tricks for the experiments in order to ensure the accuracy of solutions. Our experiments are based on the implementations on the Causal-learn package `https://github.com/py-why/causal-learn` (Zheng et al., 2023).

- DGES (ours): The first phase of our method is to run GES, as introduced above. The exact search in the third phase we incorporate is the A* as mentioned above. During the second stage, when we aim to detect the deterministic clusters and checking whether a variable can be perfectly represented by some others, we set that if the term $\|\Sigma_u\|_{HS}^2 < 1e{-}3$, although theoretically the value should exactly be zero. Meanwhile, the regularization parameter for the kernel ridge regression is set to $1e{-}10$.

### A4.2   EVALUATION ON TWO DCs

Figure 4 in the main paper presents the simulated results focused on graphs containing just a single deterministic constraint (DC). In contrast, Figure A3 in the Appendix offers insights into scenarios involving two DCs, even allowing for the possibility of overlapping variables. An evident

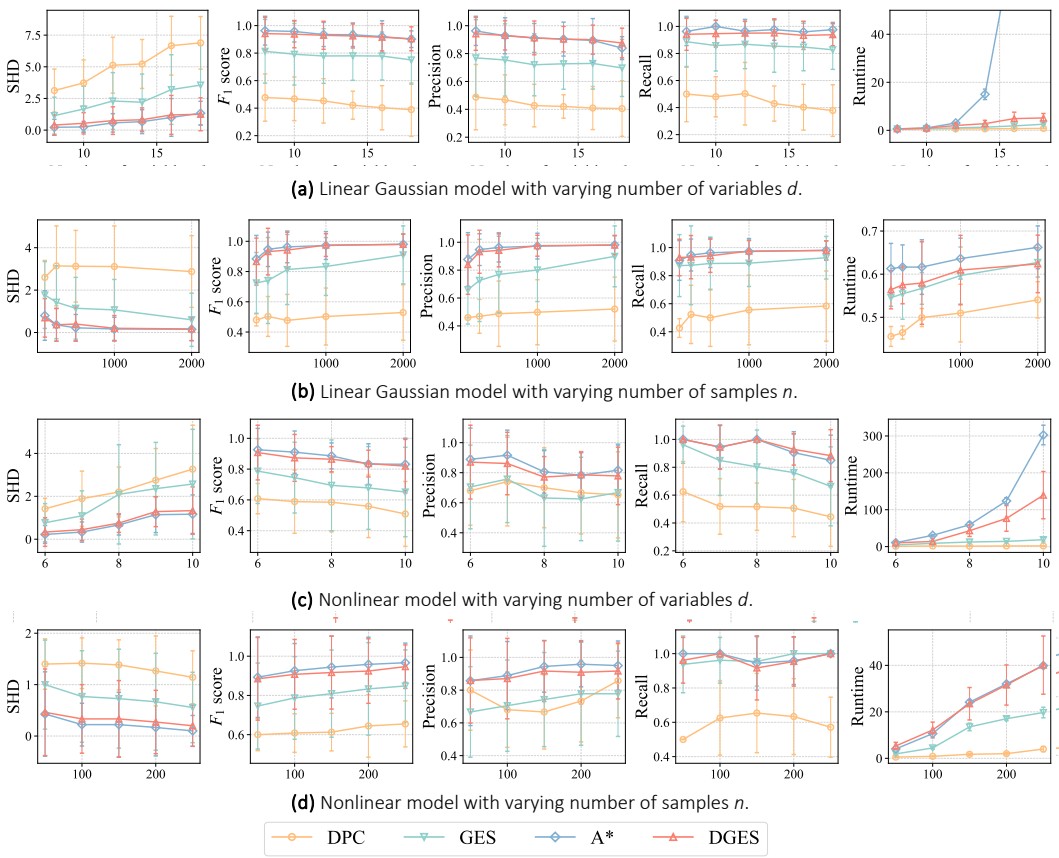

**(a)** Linear Gaussian model with varying number of variables $d$.

**(b)** Linear Gaussian model with varying number of samples $n$.

**(c)** Nonlinear model with varying number of variables $d$.

**(d)** Nonlinear model with varying number of samples $n$.

Figure A3: Simulated results on graph with two deterministic clusters. We evaluate different functional causal models on varying number of variables and samples, respectively. For each setting, we consider SHD ($\downarrow$), $F_1$ score ($\uparrow$), precision ($\uparrow$), recall ($\uparrow$) and runtime ($\downarrow$) as evaluation criteria.

trend emerges: as the system incorporates more deterministic variables, the runtime of our proposed DGES inevitably escalates. This phenomenon can be attributed to the increased number of deterministic variables demanding detection and inclusion in Phase 3, where an exact search is performed.

It is worth noting that as the number of variables in the system increases, the runtime of A* experiences a rapid surge. In stark contrast, DGES exhibits a more stable increase in runtime, demonstrating its efficiency and suitability for both linear and nonlinear models.

The outcomes gleaned from these experiments collectively indicate that DGES exhibits competitive performance compared to established baselines. Notably, the exact method A* and our proposed DGES consistently outperform other baseline methods like Greedy Equivalence Search (GES) and PC, across a spectrum of evaluation criteria and diverse settings. It is intriguing to note that in deterministic systems, the score-based method GES consistently outperforms the constraint-based method DPC. This observation suggests that score-based approaches maintain a comprehensive perspective on causal discovery, which appears to be less susceptible to the challenges posed by deterministic relationships, unlike constraint-based methods.

### A4.3 EVALUATION ON NON-DETERMINISTIC SCENARIO

We also conducted the experiments in a standard setting, where there is no deterministic relation at all. We consider the linear Gaussian model with a varying number of variables. We evaluate the SHD, the $F_1$ score, the precision, the recall, and the runtime. In this case, we evaluate based on the whole graph.

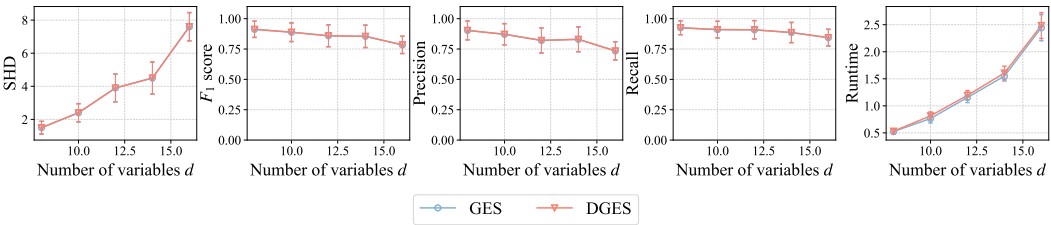

Figure A4: Results on non-deterministic scenarios on linear Gaussian model.

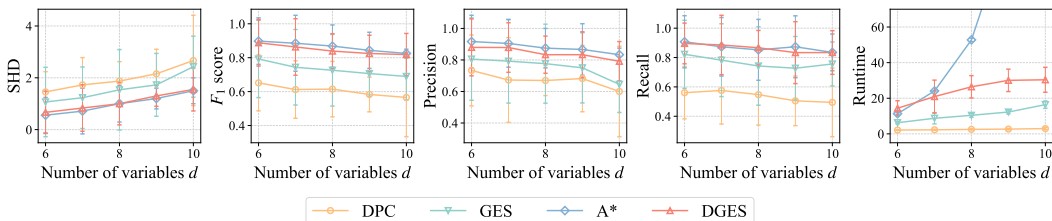

Figure A5: Results on nonlinear synthetic datasets generated by MLP.

The results have been shown in Figure A4. According to the figures, we can see that GES and our proposed DGES method present the same performance regarding the SHD, the $F_1$ score, the precision, and the recall. However, the runtime of DGES is a bit more than GES, because DGES runs 2 phases. It is understandable that when there is no deterministic relation, DGES will be reduced to GES. In Phase 2, DGES will not find any deterministic clusters, then it will terminate and output the result of GES in Phase 1.

### A4.4 EVALUATION ON MLP-BASED NONLINEAR MODEL

We also generate the synthetic nonlinear datasets by MLP to make it more general. We consider two hidden layers and each hidden layer has 100 hidden dimensions. We use Sigmiod as the activation function. All the weights are randomly generated from the uniform distribution $\mathcal{U}(0.5, 2)$. For each setting, we also run 10 different random seeds and report the mean and standard deviation.

The results have been shown in Figure A5. According to the figure, we can see that the exact method A* and our proposed DGES generally outperform the other baselines such as GES and DPC across different criteria and settings. Meanwhile, the score-based method GES presents better performance than the constraint-based method DPC in a deterministic system. The runtime of the exact search method is the highest.

### A4.5 EVALUATION ON GRASP

GRaSP (Lam et al., 2022) is a greedy relaxation of the sparsest permutation algorithm. We follow the same setting as mentioned in Section 5.1. Here we consider the linear Gaussian model with a varying number of variables, and within the generated dataset there is one deterministic cluster. We evaluate the SHD, the $F_1$ score, the precision, the recall, and the runtime. In this case, we evaluate based on only the BS part.

The results have been shown in Figure A6. According to the figure, we can see that: in general, A* and our proposed DGES still outperform other baselines. GRaSP performs slightly better than GES regarding the SHD, the $F_1$ score, the precision, and the recall. However, according to our data record, the runtime of GRaSP is a bit more than GES.

## A5 MORE DETAILS ABOUT THE REAL-WORLD EXPERIMENTS

### A5.1 MORE ANALYSIS OF DGES FOR THE REAL-WORLD DATASET

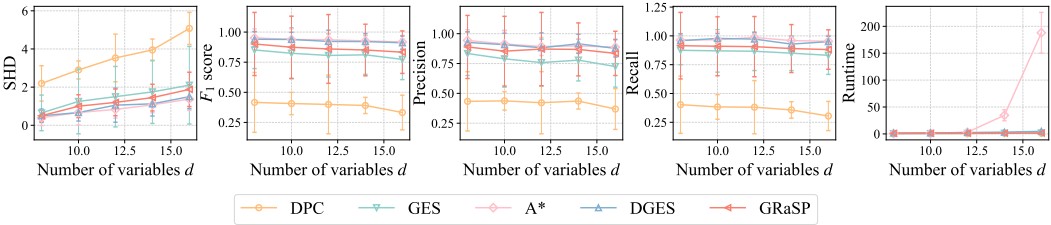

Figure A6: Results of GRaSP on synthetic linear Gaussian model.

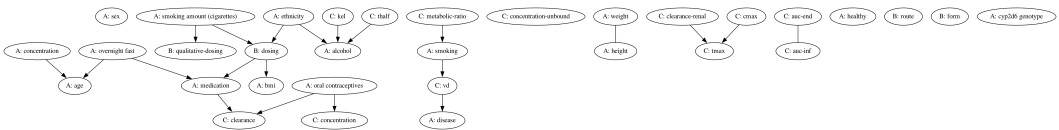

Figure A7: Results on real-world dataset with deterministic relations by PC algorithm.

Take one deterministic cluster {height, weight, BMI} as an example for further analysis. As we all know, BMI is defined as the body weight divided by the square of the body height, which composes a deterministic relation among the three variables. By applying our method, the three variables are connected like a cluster, comprising a DC. At the same time, many other non-deterministic variables still connect with at least one of the three deterministic variables, as usual. Imagine if we use a constraint-based method to deal with it, as supported by Lemma 1, there should be no edge connecting from the three variables to any others. Furthermore, according to our common knowledge, the true arrows should be {weight→BMI, height→BMI}, but now our graph just presents {weight→BMI, weight→height}. Actually this result is consistent with our theory, because currently our method cannot identify the skeleton and directions in a DC, without further assumptions, but at least we can always identify the correct deterministic clusters.

### A5.2 More analysis of PC for the Real-world Dataset

We conducted PC algorithm (Spirtes & Glymour, 1991) on the real-world dataset. We implement this method based on the Causal-learn package https://github.com/py-why/causal-learn (Zheng et al., 2023), and the significance level is set to 0.05.

The causal graph result by PC algorithm is given in Figure A7. According to the result, we can see that: compared to our proposed DGES method, PC algorithm failed to generate a reasonable causal graph.

