# OpenReview forum: "On Causal Discovery in the Presence of Deterministic Relations"
_ICLR.cc/2024/Conference — Submitted to ICLR 2024_

### Official Review · Reviewer_bELW · 2023-10-26

**Soundness:** 4 excellent
**Presentation:** 4 excellent
**Contribution:** 2 fair
**Rating:** 6
**Confidence:** 3

**Summary:**

The paper addresses the problem of causal discovery in the presence of deterministic relationships. This is a particularly challenging problem, since faithfulness is violated; that is, one cannot exploit conditional independence structures as in the classical PC or FCI approaches. The authors propose combining a score-based approach and regression to identify the Markov equivalence class. The latter is used to identify the deterministic nodes where the residual should be 0.

**Strengths:**

Nicely written and easy to follow. All important aspects are introduced clearly and explained well. The overall idea seems logical and the experimental results are convincing. I did not check the proofs in detail, but the theoretical statements make sense.

**Weaknesses:**

While the overall idea makes sense, my main concern is the lack of theoretical novelty. While the authors have some theoretical results, they are rather straightforward, and referring to them as "Theorems" may be an overstatement. For instance, it is clear that the residual is expected to be 0 if the functional relationship is perfectly represented.

**Questions:**

Although the theoretical novelty is limited, the overall approach is still valid and interesting. Some questions and remarks:

- The reference to Figure 1 in the introduction does not help in understanding the faithfulness violation with deterministic relationships. It would be clearer, for instance, to give the example of a chain X -> Y -> Z, where Y = f(X), which violates faithfulness due to the conditional independence X _||_ Y | f(X).
- The difference between Assumptions 2 and 3 could be clarified. Does Assumption 2 include the cases described in Assumption 3?
- You mention PC requires "faithfulness" but based on your definitions, it seems to require "non-deterministic faithfulness".
- Figure 1 illustrates a DC but does not clearly show why PC fails, since many factors could lead to the PC algorithm outputting graph b), even without deterministic relationships.
- The SMR assumption could be introduced in more detail, since faithfulness is common but SMR is less so.
- Theorem 4 seems overly specific. It could be more general by stating deterministic relationships will result in Var[Y | x] = 0, for any regression model. Is there a reason for using kernel regression specifically?
- In the experiments, you limit non-linear relationships to a few function classes. You could make them more arbitrary by modeling f as a neural network with random weights, to represent a random non-linear relationship.
- The examples illustrations always have edges, but your approach identifies the MEC, which has undirected parts. This distinction could be clarified.
- Does SHD refer to the whole graph or just the BS? If the latter, why not consider the whole graph?
- I like the discussion, which fairly points out some weaknesses.

**Details Of Ethics Concerns:**

I am not entirely sure if there is really a concern, but the paper uses a real-world data set (publicly available according to the authors) that seem to contain some sensitive information. I am not familiar with the data set and cannot judge if there are any concerns or not.

---

> ### Author Response · Authors · 2023-11-20
> **Responses to Reviewer bELW (1/3)**
>
> We appreciate the reviewer for the time dedicated to reviewing our paper and constructive suggestions. In light of your valuable feedback, we have **updated our manuscript**. To make a more arbitrary functional model for synthetic datasets, we have added a **new experiment** using a neural network. Please find our responses to your concerns below.
>
>
> **Q1:** "While the overall idea makes sense, my main concern is the lack of theoretical novelty. While the authors have some theoretical results, they are rather straightforward, and referring to them as "Theorems" may be an overstatement. For instance, it is clear that the residual is expected to be 0 if the functional relationship is perfectly represented."
>
> **A1:** Thanks for raising those concerns. To address your concerns, we would like to respond from the following two aspects: the theoretical novelty and contributions of our paper, and the importance of those theorems.
>
>
> **The main theoretical novelty and contributions of our paper:**
>
> - Firstly, in the paper, we intend to provide **simple and reliable** perspective to solve the issues of deterministic relations for causal discovery. We found that: under some mild assumptions, the exact search methods can output the MEC where the BS and the NDC part can be correctly identified.
>
> - Secondly, we proposed a novel method called DGES, in order to improve the efficiency of causal discovery in the presence of deterministic relations. Importantly, DGES is a **general and versatile** three-phase method, without restricted assumptions on the underlying functional causal models, in other words, it can accommodate both linear and nonlinear relationships, Gaussian and non-Gaussian data distributions, as well as continuous and discrete data types.
>
>
> **The importance of the theorems:**
>
>
> - **Theorem 3** shows that the exact search methods can be used to solve the issues of deterministic relations when SMR and some mild assumptions hold (faithfulness is not necessary anymore). Note that here we present with generalized score in a general way, where we can handle general functional causal models.
>
> - **Theorem 4** presents the techniques of how we detect the DCs (by Representation) and MinDCs (by Perfect Representation). Detecting MinDC is so fundamental that we can handle more challenging cases where there are multiple deterministic relations, even overlapping with each other. We agree that when the functional causal model is linear the theorem is straightforward, however, when we are considering the general functional causal model the scenario is more challenging, and we solve the issues by using kernel regression and evaluating the Hilbert-Schmidt norm of the infinite variance operator $\Sigma_u$.
>
> - **Theorem 5** exhibits the identifiability conditions of our proposed DGES method. Once the conditions are satisfied, the BS and NDC parts can be identifiable up to the Markov equivalent class.
>
>
> **Q2:** "The reference to Figure 1 in the introduction does not help in understanding the faithfulness violation with deterministic relationships. It would be clearer, for instance, to give the example of a chain X -> Y -> Z, where Y = f(X), which violates faithfulness due to the conditional independence X || Y | f(X)."
>
> **A2:** Thank you so much for this constructive suggestion. More details have been included in Section 1 of our manuscript, i.e., "Take the chain structure $X \rightarrow Y \rightarrow Z$ for example where $Y=f(X)$. In this case, faithfulness is violated due to the conditional independence $Y \perp Z | X$ or $f(X) \perp Z | X$."
>
>
> **Q3:** "The difference between Assumptions 2 and 3 could be clarified. Does Assumption 2 include the cases described in Assumption 3?"
>
> **A3:**  Thanks a lot for this question. We would like to clarify that: Faithfulness (Assumption) is usually used for the whole set of variables, no matter whether there is a deterministic relation or not. On the contrary, non-deterministic faithfulness (Assumption 3) is only assumed for those non-deterministic parts of the whole graph. We have included those details in Section 2 of our updated manuscript to avoid confusion.
>
> **Q4:** "You mention PC requires 'faithfulness' but based on your definitions, it seems to require 'non-deterministic faithfulness'."
>
> **A4:** Thanks for this question. We would like to clarify that: the PC algorithm requires "faithfulness" for the whole set of variables, which is why the faithfulness will be violated in the example of Figure 1. However, another modified PC algorithm (called DPC) assumes "non-deterministic faithfulness", as mentioned in Section 1 and Section 2.

---

> > ### Author Response · Authors · 2023-11-20
> > **Responses to Reviewer bELW (2/3)**
> >
> > **Q5:** "Figure 1 illustrates a DC but does not clearly show why PC fails, since many factors could lead to the PC algorithm outputting graph b), even without deterministic relationships."
> >
> > **A5:** Thanks for the question. The **key rule of the PC algorithm** is that: as long as we can find at least one conditional set or an empty set, so that two variables are independent or conditionally independent, then the edge between these two variables will be removed. In the example of Figure 1(b), we find that $V_4 \perp  V_3|\{V_1,V_2\}$, therefore the edge between $V_4$ and $V_3$ will be removed, which violates the faithfulness assumption. We have included this detail in Section 2 of our updated manuscript.
> >
> >
> > **Q6:** "The SMR assumption could be introduced in more detail, since faithfulness is common but SMR is less so."
> >
> > **A6:** Thanks a lot for this great point. The idea behind the SMR is to find the most parsimonious or sparsest graphical representation that still captures the essential conditional independence relationships in the data. The term "sparsest" refers to the minimal number of edges in the graphical model. We have included more details about SMR assumption in Section 2 of our updated manuscript.
> >
> >
> > **Q7:** "Theorem 4 seems overly specific. It could be more general by stating deterministic relationships will result in Var[Y | x] = 0, for any regression model. Is there a reason for using kernel regression specifically?"
> >
> > **A7:** We appreciate this great point. The reason why we use kernel regression is that: we are considering the **general functional causal form**, particularly, this theorem can be used for both linear and nonlinear functional relationships, Gaussian and non-Gaussian data distributions, which is in alignment with the general goal of our proposed method.
> >
> > For more details, inspired by [1], the functions $\phi_{\mathcal{X}}$ and $F(\cdot)$ that we use are all in the **infinite Hilbert spaces**, and we evaluate the representation with the Hilbert-Schmidt norm of the variance operator $\Sigma_u$ in infinite dimension. In this case, we can exhibit a general functional causal form.
> >
> > We have included the discussions above in Appendix 1 of our updated manuscript.
> >
> >
> > **Q8:** "In the experiments, you limit non-linear relationships to a few function classes. You could make them more arbitrary by modeling f as a neural network with random weights, to represent a random non-linear relationship."
> >
> > **A8:** Thanks for this insightful question. Actually, We just follow previous work [2] and choose this mixture of functions. In light of your comment, we have added a new set of experiments to generate the synthetic datasets with a neural network and included the results in Appendix A4.4. The results show that our proposed method can also work well under this generation scenario.
> >
> >
> > **Q9:** "The examples illustrations always have edges, but your approach identifies the MEC, which has undirected parts. This distinction could be clarified."
> >
> > **A9:** Thanks for your comments. In light of your suggestions, we have included more details for clarification in Section 3 of the updated manuscript.
> >
> >
> > **Q10:** "Does SHD refer to the whole graph or just the BS? If the latter, why not consider the whole graph?"
> >
> > **A10:** We appreciate this great question. As mentioned in Section 5.1, we focus on the BS part for all the evaluation metrics including SHD. The reason is that our method cannot identify the skeleton and directions in the DC part so far, and below are more details.
> >
> > - In order to identify the skeleton and directions in the DCs, we usually need strong assumptions on the underlying functional causal model, i.e., Yang et al. [1] assumed linear Non-Gaussian model and utilized causal asymmetry to identify the graph structure with deterministic relations. However, those extra assumptions are not in alignment with our initial goal of this paper, to provide a general and versatile method to deal with deterministic relations for causal discovery. Currently, our method makes no assumption on the underlying functional causal models, in other words, it can accommodate both linear and nonlinear relationships, Gaussian and non-Gaussian data distributions, as well as continuous and discrete data types.
> >
> > We have included the discussions above in Section 6 and Appendix 1 of our updated manuscript.
> >
> >
> > **Q11:** "I like the discussion, which fairly points out some weaknesses."
> >
> > **A11:** Thanks a lot for your appreciation, and we are deeply encouraged by your kind words.
> >
> > ---
> >
> > We sincerely thank you once again for the valuable and constructive suggestions. We hope you will find that our responses, along with updated manuscripts, have properly addressed your concerns. Please kindly let us know if there are any further questions or comments.

---

> > > ### Author Response · Authors · 2023-11-20
> > > **Responses to Reviewer bELW (3/3)**
> > >
> > > **References:**
> > >
> > > [1] Zhang, Peters, et al. "Kernel-based conditional independence test and application in causal discovery." UAI, 2012.
> > >
> > > [2] Huang, Zhang, et al. "Causal discovery from heterogeneous/nonstationary data."  Journal of Machine Learning Research, 2020.

---

### Official Review · Reviewer_fVdc · 2023-10-28

**Soundness:** 3 good
**Presentation:** 3 good
**Contribution:** 3 good
**Rating:** 8
**Confidence:** 3

**Summary:**

The surprising result from the paper is that exact search methods can be used to address issues from deterministic relations in causal discovery. The authors show how the presence of deterministic functionals can lead to violation of faithfulness. Particularly, it proposes a variant of score-based causal discovery method known as DGES to increase efficiency of using exact search methods by first detecting potential deterministic clusters and their neighbors.

**Strengths:**

* The paper is well-organized and the problem is well-motivated. The claims are sound and strongly supported by experiments.
* The authors show that faithfulness fails due to deterministic relations by Lemma 1 and they also extend previous results about how exact score-based methods can work in non-linear case, by Theorem 3, with the sparest Markov assumption, which is strictly weaker than faithfulness.
* It provides results (Theorem 4) on how to detect a set of deterministic variables and their neigbors from a Markov equivalence class and use that to increase the efficiency of using exact search methods on the DCs and their neighbors.

**Weaknesses:**

* In the proof of Theorem 3, there is a reference missing with question mark.
* As acknowledged by the authors that the edge directions for deterministic clusters cannot be determined without further assumptions on the functional relationships.
* Using exact search methods can be computationally expensive.

**Questions:**

* Will algorithms that rely on SMR assumption outperform GES-based methods when deterministic relations are present? I am curious about how well GRaSP[1] will perform in the experiment given the result from Theorem 5. Can the authors include that in the experiment?
* How can one distinguish the violation of non-deterministic faithfulness and the conditional independence relations between deterministic variable and some non-deterministic variables in practice? Does that mean need to domain knowledge to determine whether they should use DGES or they should use DGES in general over GES? Can the authors show some experimental results on cases when there are no deterministic relations to see if the proposed algorithm is at least as good as GES?
* In Theorem 5, the authors show that BS and NDC are identifiable up to Markov equivalence class, but it is not clear to me what the characterization of Markov equivalence class is in the context of having deterministic relations in the graph as a whole. From Lemma 1, we know that there are CI statements that cannot be read-off from the graph, shouldn't there be a different characterization of Markov equivalence class?

Reference:

[1] Lam, W. Y., Andrews, B., & Ramsey, J. (2022, February). Greedy Relaxations of the Sparsest Permutation Algorithm. In The 38th Conference on Uncertainty in Artificial Intelligence.

---

> ### Author Response · Authors · 2023-11-20
> **Responses to Reviewer fVdc (1/2)**
>
> We appreciate the reviewer for the time dedicated to reviewing our paper, constructive suggestions, and encouraging feedback. We have carefully **modified the manuscript** in light of your detailed suggestions. Additionally, we have conducted **a new set of experiments**. Please find the responses to all your comments below.
>
> **Q1:** "In the proof of Theorem 3, there is a reference missing with question mark."
>
> **A1:** Thanks for your careful reading and raising this issue. We have filled in the reference in Appendix A3.2.
>
> **Q2:** "As acknowledged by the authors the edge directions for deterministic clusters cannot be determined without further assumptions on the functional relationships."
>
> **A2:** Thank you so much for your comment. We would like to respond to the following points.
>
> - In order to identify the skeleton and directions in the deterministic clusters (DCs), we usually need strong assumptions on the underlying functional causal model, i.e., Yang et al. [1] assumed linear Non-Gaussian model and utilized causal asymmetry to identify the graph structure with deterministic relations. However, those extra assumptions are not in alignment with our initial goal of this paper, to provide a general and versatile method to deal with deterministic relations for causal discovery. Currently, our method makes no assumption on the underlying functional causal models, in other words, it can accommodate both linear and nonlinear relationships, Gaussian and non-Gaussian data distributions, as well as continuous and discrete data types.
>
> - Although our method cannot identify the skeleton and directions in the DCs, we proposed strategies to exactly find out which set of variables have deterministic relations, as shown in Section 3.3.
>
> - Nevertheless, in light of your comments, we are glad to continue studying the identification in DCs with mild assumptions as future work.
>
> We have included those details discussed above in Section 6 and Appendix 1 of our updated manuscript.
>
>
> **Q3:** "Using exact search methods can be computationally expensive."
>
> **A3:** Thanks a lot for the comment. We agree that the exact search methods are computationally expensive, which motivates us to propose our DGES method where we only need to run the exact search on a small set of variables in Phase 3, instead of all variables.
>
> **Q4:** "Will algorithms that rely on SMR assumption outperform GES-based methods when deterministic relations are present? I am curious about how well GRaSP[1] will perform in the experiment given the result from Theorem 5. Can the authors include that in the experiment?"
>
> **A4:** Thanks for the great suggestion. In general, the exact search-based methods (with SMR assumption) will outperform GES-based methods (with faithfulness assumption). For example, Figure 2 shows the case where GES might be problematic when there is a deterministic relation, however, the exact search methods can solve this issue.
>
> In light of your suggestion, we have included the experiment in Appendix A4.5 to evaluate how GRaSP [2] performs in the presence of deterministic relations. According to the result, we can see that: In general, GRaSP performs slightly better than GES regarding the SHD, the $F_1$ score, the precision, and the recall. However, the runtime of GRaSP is a bit more than GES. Moreover, A* and our proposed DGES still outperform other baselines, including GRaSP. The reason could be that GRaSP is a relaxation-based method which may lose some solution accuracy compared to the exact method.

---

> > ### Author Response · Authors · 2023-11-20
> > **Responses to Reviewer fVdc (2/2)**
> >
> > **Q5:** "How can one distinguish the violation of non-deterministic faithfulness and the conditional independence relations between deterministic variable and some non-deterministic variables in practice? Does that mean need to domain knowledge to determine whether they should use DGES or they should use DGES in general over GES? Can the authors show some experimental results on cases when there are no deterministic relations to see if the proposed algorithm is at least as good as GES?"
> >
> > **A5:** We appreciate your insightful comment.
> >
> > - To distinguish the violation of non-deterministic faithfulness and the conditional independence relations between deterministic and non-deterministic variables, we need to rely on the result of detected DCs as shown in Section 3.3, in other words, we have to know firstly which sets of variables are DCs and which set of variables is NDC.
> >
> > - Here we do not necessarily need the domain knowledge. Instead, if we can detect some DCs in Phase 2, then we will continue Phase 3 for DGES; Otherwise, we will stop in Phase 2, and just return the result by Phase 1 using GES.
> >
> > - In light of your suggestion, we have included the experiment in Appendix A4.3 where there is no deterministic relation. According to the result, we can see that GES and our proposed DGES method present the same performance regarding the SHD, the $F_1$ score, the precision, and the recall. However, the runtime of DGES is a bit more than GES, because DGES runs 2 phases. It is understandable that when there is no deterministic relation, DGES will be reduced to GES. In Phase 2, DGES will not find any deterministic clusters, then it will terminate and output the result of GES in Phase 1.
> >
> > **Q6:** "In Theorem 5, the authors show that BS and NDC are identifiable up to Markov equivalence class, but it is not clear to me what the characterization of Markov equivalence class is in the context of having deterministic relations in the graph as a whole. From Lemma 1, we know that there are CI statements that cannot be read-off from the graph, shouldn't there be a different characterization of Markov equivalence class?"
> >
> > **A6:** Thank you for this insightful question. Regarding only the variables involved in BS and NDC (that is how Theorem 5 claimed), the characterization of Markov equivalence class (MEC) is still the same in the context of having deterministic relations or not having such relations.
> >
> > However, if we consider the whole graph, i.e., all of the variables in DCs are also involved, the characterization of the Markov equivalence class should be different. As shown in the example of Lemma 1 and Figure 1, there will be "spurious CI statements" caused by the deterministic relations, therefore, we need to remove those "spurious CI statements" for the new characterization of MEC.
> >
> > We have included those details discussed above in Appendix 1 of our updated manuscript.
> >
> >
> > Thank you again for your constructive comments, which are really helpful in improving the quality of the manuscript. Meanwhile, thanks a lot for the appreciation of our work, and we are really encouraged by it. We hope our responses and the modified manuscript can adequately address the remaining concerns. Please let us know if there are any further questions.
> >
> >
> > ---
> > **References:**
> >
> > [1] Yang, Nafea, et al. "Causal discovery in linear structural causal models with deterministic relations." Conference on Causal Learning and Reasoning, 2022.
> >
> > [2] Lam, Andrews, and Ramsey. "Greedy Relaxations of the Sparsest Permutation Algorithm." The Conference on Uncertainty in Artificial Intelligence. 2022.

---

> > > ### Comment · Reviewer_fVdc · 2023-11-22
> > > **Appreciate for the authors response**
> > >
> > > I appreciate for the response. I have also read the authors' response to other questions. I think all the questions are well-addressed and I decide to raise my score.

---

> > > > ### Author Response · Authors · 2023-11-22
> > > > **Thank you so much for checking the response and updating your recommendation**
> > > >
> > > > We are glad that our response and the updated manuscript have well addressed your questions. Thank you again for your valuable time and constructive suggestions!

---

### Official Review · Reviewer_qsqn · 2023-10-31

**Soundness:** 3 good
**Presentation:** 3 good
**Contribution:** 2 fair
**Rating:** 6
**Confidence:** 4

**Summary:**

In causal discovery, one of the challenges is the presence of deterministic relations, which violates the common assumption of independent noise. This paper proposes a framework called determinism-aware greedy equivalent search (DGES) that can detect the deterministic clusters from data using score-based method, encompassing both linear and nonlinear models. Theoretical guarantees and empirical experiments are provided to support the novel framework.

**Strengths:**

The paper is well-organized, and the figures are illustrative. The existence of deterministic relations is common in real-world applications and results in biased or even false inference about the data structure. The novel framework DGES alleviates the problem by separate the interested variables into deterministic clusters and non-deterministic clusters and specifically focus on the DCs. DGES is also flexible regarding the linear and non-linear models, as well as the discrete and continuous data.

**Weaknesses:**

First, the authors claim that constraint-based methods suffer from the deterministic relations and briefly state the reason on page 3. It would be more convincing if solid theorems or experiments are provided to show the failure of constraint-based methods. In addition, how serious is the assumption violation here? Do all constraint-based methods failed or some adjustments would fix this problem?
Second, as the authors mentioned themselves, the DGES framework can not identify the skeleton and directions in the DCs, which negatively influence the utilization of DGES compared with other mature developed methods.
Thirdly, I doubt could DGES identify all possible DCs or just part of the deterministic relations? If the answer is no, how could we deal with the missing ones?

**Questions:**

1. As for the DCs, could there be case that two DCs connected by a non-deterministic edge? If so, how the DGES framework handle this kind of situation?
2. For the real-world example in 5.2, a comparison between the result using DGES and the results using other classic methods like PC would be interesting. It would be an illustrative way to show how proposed framework differently handle the deterministic relations.
3. Given that the skeleton and directions in DCs are not accurate, would the NDCs and BSs suffer from the same problem?

---

> ### Author Response · Authors · 2023-11-20
> **Responses to Reviewer qsqn (1/2)**
>
> We appreciate the reviewer for the time dedicated to reviewing our paper and constructive suggestions. With the help of this valuable feedback, we believe that our manuscript could be improved a lot. We have added the **new experiment** for evaluating how PC performs in the real-world dataset, and **updated our manuscript** according to your detailed suggestions. Please find the point-by-point responses below.
>
> **Q1**: "First, the authors claim that constraint-based methods suffer from deterministic relations and briefly state the reason on page 3. It would be more convincing if solid theorems or experiments were provided to show the failure of constraint-based methods. In addition, how serious is the assumption violation here? Do all constraint-based methods fail or some adjustments would fix this problem?"
>
> **A1**: We appreciate your constructive comments. We would like to address your concerns one by one as follows.
>
> - To show the failure of the constraint-based method, we have provided Lemma 1. Due to the space limit, we just gave the basic idea of the proof in the main paper and put the detailed theoretical proof in Appendix A3.1.
>
> - The faithfulness assumption violation is quite serious. The **key rule of constraint-based method** (e.g., PC algorithm) is that: as long as we can find at least one conditional set or an empty set, so that two variables are independent or conditionally independent, then the edge between these two variables in the graph will be removed. Take Figure 1 as the example, where $V_3$ is deterministically determined by $V_1$ and $V_2$ while $V_4$ is dependent on $V_3$. According to Lemma 1, $V_4\perp V_3|\{V_1,V_2\}$ always holds true. Therefore, $V_4$ and $V_3$ will never have an edge in the graph using the constraint-based method, although there is an edge between them in the true graph.
>
> - If following the original key rule, most constraint-based methods will fail. However, there are some papers attempting to fix this problem by incorporating some adjustments. For example, Glymour [1] proposed a heuristic procedure to learn the causal graph in a deterministic system, called **DPC**. When testing $V_1 \perp V_2|S$ in the presence of deterministic relations, DPC will remove those variables from the conditional set $S$ if they are strongly correlated to $V_1$ or $V_2$, to avoid violating faithfulness.
>
> - In our original manuscript, we briefly introduced the idea of DPC in Section 1, and we also provided the experiments with DPC in Section 5 and Figure 3. However, the performance of DPC is not so promising.
>
> We have included those details discussed above in Section 2.2 of our updated manuscript.
>
> **Q2**: “Second, as the authors mentioned themselves, the DGES framework can not identify the skeleton and directions in the DCs, which negatively influence the utilization of DGES compared with other mature developed methods.“
>
> **A2**: Thanks a lot for the insightful comments. We would like to address your concerns on the following points.
>
> - In order to identify the skeleton and directions in the DCs, we usually need strong assumptions on the underlying functional causal model, i.e., Yang et al. [1] assumed linear Non-Gaussian model and utilized causal asymmetry to identify the graph structure with deterministic relations. However, those extra assumptions are not in alignment with our initial goal of this paper, to provide a general and versatile method to deal with deterministic relations for causal discovery. Currently, our method **makes no assumption on the underlying functional causal models**, in other words, it can accommodate both linear and nonlinear relationships, Gaussian and non-Gaussian data distributions, as well as continuous and discrete data types.
>
> - Although our method cannot identify the skeleton and directions in the DCs, we proposed the **DC detection strategies** to exactly find out which set of variables have deterministic relations, as shown in Section 3.3.
>
> - Nevertheless, in light of your comments, we are glad to continue studying the identification in DCs with mild assumptions as future work.
>
> We have included those details discussed above in Section 6 and Appendix 1 of our updated manuscript.
>
> **Q3**: "Thirdly, I doubt could DGES identify all possible DCs or just part of the deterministic relations. If the answer is no, how could we deal with the missing ones?"
>
> **A3**: Thanks for raising this great point. With Theorem 4 (Representation and Perfect Representation), DGES can identify all possible DCs. Particularly, with the "Perfect Representation" theorem, we can identify all the minimal DCs (MinDCs). When combining all the variables in all the MinDCs, we can obtain the whole set of variables that are in the presence of deterministic relations.

---

> > ### Author Response · Authors · 2023-11-20
> > **Responses to Reviewer qsqn (2/2)**
> >
> > **Q4**:  "As for the DCs, could there be a case that two DCs connected by a non-deterministic edge? If so, how the DGES framework handle this kind of situation?"
> >
> > **A4**:  Thanks for pointing out this question. Yes, there could be a case where two DCs are connected by a non-deterministic edge. For example, there are 5 variables, $V_3=f(V_1,V_2)$, $V_5=f(V_4)$, and $V_4=f(V_3,\epsilon)$ ({$V_1,V_2,V_3$} and {$V_4,V_5$} are two DCs, and the edge between $V_3$ and $V_4$ are non-deterministic). In this case, our method will run GES in Phase 1 to get an initial graph. In Phase 2, our method can detect two separate DCs (which are also MinDCs, i.e., {$V_1,V_2,V_3$} and {$V_4,V_5$}). In Phase 3, an exact search method will run based on the detected DCs and their neighbors. Fortunately, in this case, the second condition of partial identifiability is satisfied, because the parent of $V_4$ has just 1 variable (|${PA}_4$|=|$V_3$|=1), and the minimal DC where $V_3$ belongs to has a cardinality of 3 (|MinDC|=3), therefore, |${PA}_4$|<|MinDC|-1. Once the first condition is satisfied, partial identifiability is guaranteed (i.e., the BS and NDC parts are identifiable up to MEC).
> >
> > Furthermore, in our experiments, we even considered a more challenging case for the synthetic dataset, where there are two DCs that may have overlapping variables between them. Please refer to Appendix A4.2 and Figure A3 for more details. The results showed that our proposed method can also work well under this challenging case.
> >
> > **Q5**: "For the real-world example in 5.2, a comparison between the result using DGES and the results using other classic methods like PC would be interesting. It would be an illustrative way to show how the proposed framework differently handles the deterministic relations."
> >
> > **A5**: Thanks for this interesting suggestion. We have included the experiment in Appendix A5, using PC to solve the real-world dataset. The resulting graph showed that the PC algorithm failed to get a reasonable graph, especially many variables or sets of variables are isolated from other variables, which can be caused by the deterministic relations.
> >
> >
> > **Q6**: "Given that the skeleton and directions in DCs are not accurate, would the NDCs and BSs suffer from the same problem?"
> >
> > **A6**: Thank you so much for the question. The NDC and BS will not suffer from the problem. As long as the Markov assumption and Non-deterministic Faithfulness assumption are satisfied, the NDC part can be identified. Moreover, as long as the SMR assumption and the condition ($\operatorname{PA}_{V_4}$|<|MinDC|-1) are satisfied, the BS part can also be identified.
> >
> > Even though our method cannot identify the skeleton and directions within the DCs, we can still detect which set of variables have deterministic relations by Phase 2 (as shown in Section 3.3), in other words, we can identify the true DCs.
> >
> >
> >
> > We sincerely thank you once again for the constructive and insightful suggestions. We hope you will find that our responses, along with updated manuscripts and new experiments, have properly addressed your concerns. Please kindly let us know if there are any further questions or comments.
> >
> >
> > ***
> > **References:**
> >
> > [1] Clark Glymour. "Learning the structure of deterministic systems." Causal learning: Psychology, philosophy, and computation, 2007.

---

### Official Review · Reviewer_dg4D · 2023-11-05

**Soundness:** 4 excellent
**Presentation:** 4 excellent
**Contribution:** 2 fair
**Rating:** 5
**Confidence:** 4

**Summary:**

The authors propose a version of GES that, under specific assumptions, can learn correct Markov equivalence classes for systems that include deterministic dependencies.

**Strengths:**

The paper is well-written. It deals with a reasonably important problem. It clearly locates its contribution within the large context of the existing literature. It provides both theoretical and empirical evidence regarding its key claims.

**Weaknesses:**

Unless I'm missing something, the authors results are implied by the following: (1) Due to its scoring function (which penalizes unnecessary edges), GES chooses the MEC for the DC with fewest edges; (2) When the SMR assumption is satisfied, the correct MEC is guaranteed to be the one with fewest edges; and (3) GES is known to find the MEC for the NDC.  Viewed this way, the results of the paper follow directly from what is already known about GES and the SMR assumption, and thus the contribution is fairly small. If this is incorrect, the authors should explain why in the paper. If the above is correct but there is more to it than that, the authors should clearly state the above intuition and provide more discussion of what is missing from that story.

Minor issues:

In Section 1, the authors state that some methods "suffer from identifiablity guarantees." It is unclear what this means.

In the abstract and Section 1, the authors state that they "excitingly find" that certain methods can address the issues of deterministic relations. It is enough to state the finding, without excitement.

**Questions:**

The authors show cases in which DC variables cause each other, and when DC variables cause NDC variables (e.g., Figure 2). However, they do not explicitly discuss cases in which an NDC variable causes a DC variable. That is, variables caused by (latent) noise (i.e., NDC variables) could cause variables without latent noise (i.e., DC variables). Did I miss a discussion, is this case just implied, or is this an assumption of the approach?

---

> ### Author Response · Authors · 2023-11-20
> **Responses to Reviewer dg4D (1/2)**
>
> We greatly appreciate the reviewer’s time, encouraging comments, and constructive suggestions. We have carefully **modified the manuscript** according to your detailed suggestions. Please find the responses to all your comments below. Q1-Q4 correspond to the points in “Weaknesses”, while Q5 corresponds to the point in “Questions”.
>
> **Q1**: “Unless I'm missing something, the author's results are implied by the following: (1) Due to its scoring function (which penalizes unnecessary edges), GES chooses the MEC for the DC with the fewest edges; (2) When the SMR assumption is satisfied, the correct MEC is guaranteed to be the one with fewest edges; and (3) GES is known to find the MEC for the NDC.”
>
> **A1**: We sincerely appreciate your summarization. The third point is true, while the first two points are not exactly true. Thus, we would like to clarify the first two points as follows: (1) With a proper score function (e.g., generalized score) and some mild assumptions (e.g., SMR), **the exact search methods** can output a MEC where **only the NDC and BS parts** can be correctly identified with fewest edges. GES could be problematic regarding the BS part (i.e., there might be more edges than the true graph, one example is given in Figure 2(b)). (2) With both the non-deterministic assumption (for the NDC part) and the SMR assumption (for the BS part), the correct MEC (regarding only the NDC and BS parts) is guaranteed to be the one with the fewest edges.
>
> **Q2**: "Viewed this way, the results of the paper follow directly from what is already known about GES and the SMR assumption, and thus the contribution is fairly small. If this is incorrect, the authors should explain why in the paper. If the above is correct but there is more to it than that, the authors should clearly state the above intuition and provide more discussion of what is missing from that story."
>
> **A2**: Thanks for your constructive suggestion. We would like to clarify our contributions as follows.
>
> - Firstly, in the paper, we intend to provide a **simple and reliable** view to deal with the issues of deterministic relations for causal discovery. We found that: under some mild assumptions, the exact search methods can naturally used to solve the deterministic issues and output the MEC where both the BS part and the NDC part, can be correctly identified.
>
> - Secondly, we proposed a novel method called DGES, in order to improve the efficiency of causal discovery in the presence of deterministic relations. Importantly, DGES is a **general and versatile** three-phase method, with **no restricted assumption** on the underlying functional causal models, in other words, it can accommodate both linear and nonlinear relationships, Gaussian and non-Gaussian data distributions, as well as continuous and discrete data types. Specifically, **theoretical analysis and empirical evidence** were provided to support our claim.
>     - Theoretically, we provided the **identifiability conditions** for the BS part, besides the NDC part benefiting from GES.
>     - Empirically, we conducted extensive experiments on both simulated and real-world datasets to show the efficacy of our proposed method.
>
> (If you are interested, here we would like to explain why our method currently cannot identify the edges in the DC part. To achieve that goal, we usually need strong assumptions on the underlying functional causal model, i.e., Yang et al. [1] assumed the linear non-Gaussian model. However, those assumptions are not in alignment with our goal of a general method. That is why currently our method cannot identify the skeleton and directions in the DC part. However, fortunately, we can exactly find out which set of variables are in the DCs using some **DC detection strategies**, as shown in Section 3.3.)
>
> We have included those details discussed above in Section 1 of our updated manuscript.
>
>
> **Q3**: "In Section 1, the authors state that some methods 'suffer from identifiability guarantees.' It is unclear what this means."
>
> **A3**: Thanks a lot for raising this concern. By "suffer from identifiability guarantees", we mean that there is no identifiability guarantee in those related works. In light of your comments, we have updated the expression to avoid confusion in the manuscript.
>
>
> **Q4**: "In the abstract and Section 1, the authors state that they 'excitingly find' that certain methods can address the issues of deterministic relations. It is enough to state the finding, without excitement."
>
> **A4**: Thank you so much for your careful reading and helpful suggestions. We have updated our expressions in the abstract and Section 1 of the manuscript, respectively.

---

> > ### Author Response · Authors · 2023-11-20
> > **Responses to Reviewer dg4D (2/2)**
> >
> > **Q5**: "The authors show cases in which DC variables cause each other, and when DC variables cause NDC variables (e.g., Figure 2). However, they do not explicitly discuss cases in which an NDC variable causes a DC variable. That is, variables caused by (latent) noise (i.e., NDC variables) could cause variables without latent noise (i.e., DC variables). Did I miss a discussion, is this case just implied, or is this an assumption of the approach?"
> >
> > **A5**: Thanks a lot for your insightful question. In the case where the NDC variable causes a DC variable, the edge from the NDC variable to the DC variable can be correctly identified by GES, therefore, we did not specify this case. Below we give more details.
> >
> > - Let's consider this example. Three variables compose a DC ($V_1, V_2, V_3$, and $V_1=V_2+V_3$), and denote another variable from NDC as $V_4$. In this case, there must not be an edge from $V_4$ to $V_1$, because $V_4$ will be in DC rather than NDC, if so. Then, if $V_4$ causes $V_2$, there will be definitely an edge between them by GES, because $V_4$ is clearly dependent on $V_2$ and theoretically GES can capture this dependence based on the local consistency.
> >
> > Here, we also would like to share more details about why we only consider the case where DC variables cause NDC variables, e.g., the example in Figure 2 (below are more details). The reason is that: in this case, GES can get trapped in the local solution, where one NDC variable is very likely to be caused by more DC variables, which is incorrect. Therefore, we need to run an exact search under the SMR assumption in Phase 3, in order to identify the true BS part.
> >
> > - Take Figure 2 as an example, where the true edges related to $V_6$ include $V_3\rightarrow V_6$ and $V_4\rightarrow V_6$. However, during the forward phase of GES, it is very likely that the edge $V_1 \rightarrow V_6$ can be added in the beginning. Then the edges $V_2 \rightarrow V_6$ and $V_4 \rightarrow V_6$ are added subsequently. During the backward phase of GES, the edge $V_1 \rightarrow V_6$ will not be deleted, because $V_1$ also contains information from $V_3$, in other words, $V_6$ is represented by $V_1, V_2$ and $V_4$ by GES, which contains more edges than the ground-true. Therefore, in this case, GES can be problematic, and we need an exact search under the SMR assumption for post-processing, in order to correctly identify the BS part.
> >
> > We have included those details discussed above in Section 3 and Appendix 1 of our updated manuscript.
> >
> >
> > Thank you again for all these constructive comments, which are really helpful in improving the quality of the manuscript. We hope our responses and the modified manuscript can adequately address the concerns. Please let us know if there are any further questions.
> >
> >
> > ---
> > **References:**
> >
> > [1] Yang, Nafea, et al. "Causal discovery in linear structural causal models with deterministic relations." Conference on Causal Learning and Reasoning, 2022.

---

### Author Response · Authors · 2023-11-20
**Responses to All Reviewers**

Thanks a lot for the constructive reviews from all the reviewers. We are encouraged by some of your positive comments and also inspired by your insightful suggestions which really help in improving the quality of our manuscript. We here provide more details regarding where and how we have modified our manuscript.

- To ALL reviewers:
    - More details have been included in Sections 1, 2, 3, and 6.
    - We have moved the analysis of the real-world dataset to Appendix A5.1 to save more space.
    - To improve the presentation and avoid confusion, we have added the discussions of some insightful questions raised by the reviewers in Appendix A1.
    - More experimental results and analysis are provided in Appendix A4 and A5.

- To Reviewer qsqn:
    - We have added a new experiment to evaluate how PC performs with the real-world dataset, as shown in Appendix A5.2.

- To Reviewer fVdc:
    - We have added a new experiment to evaluate how DGES and GES perform in the synthetic dataset when there is no deterministic relation, as shown in Appendix A4.3.
    - We have added a new experiment to evaluate how GRaSP performs with the synthetic dataset, as shown in Appendix A4.5.

- To Reviewer bELW:
    - We have added a new experiment where we use a neural network to generate the synthetic dataset, as shown in Appendix A4.4.

---

### Meta-Review · Area_Chair_jetS · 2023-12-05

**Metareview:**

The authors propose a modified version of GES to learn Markov equivalence from data in the presence of deterministic relations. The reviewers acknowledge the importance of the problem, although there was some disagreement in terms of technical novelty and contribution.

Based on my own reading, I am a bit concerned on some of the statements. Combined with the reviewers' concerns around clearly delineating the contributions of the paper I believe the authors should carefully revise the paper based on reviewer comments. Specifically, it is not clear to me why Lemma 1 is not obvious: In any algebraic equality, given all but one variable, the remaining variable is a constant and any constant is independent from anything else by definition. The authors have a proof that goes through regression which confuses me since regression should not be part of the proof here without more assumptions such as Gaussianity. Thus, I think the authors should carefully revise the claims and proofs in the paper. This observation is consistent with one of the reviewer remarks "While the authors have some theoretical results, they are rather straightforward, and referring to them as "Theorems" may be an overstatement.".

**Justification For Why Not Higher Score:**

The novelty and technical contribution are seen as borderline by many of the reviewers. This itself is not grounds for rejection but combined with unclear writing around the papers' contributions make it a borderline where I think the paper would significantly benefit from a revision.

**Justification For Why Not Lower Score:**

N/A

---

### Decision · Program_Chairs · 2024-01-16

Reject